# Air pollution disparities and equality assessments of US national decarbonization strategies

Teagan Goforth [1] ✉ & Destenie Nock [1,2] ✉

Energy transitions and decarbonization require rapid changes to a nation's electricity generation mix. There are many feasible decarbonization pathways for the electricity sector, yet there is vast uncertainty about how these pathways will advance or derail the nation's energy equality goals. We present a framework for investigating how decarbonization pathways, driven by a least-cost paradigm, will impact air pollution inequality across vulnerable groups (e.g., low-income, minorities) in the US. We find that if no decarbonization policies are implemented, Black and high-poverty communities may be burdened with 0.19–0.22 μg/m$^3$ higher PM$_{2.5}$ concentrations than the national average during the energy transition. National mandates requiring more than 80% deployment of renewable or low-carbon technologies achieve equality of air pollution concentrations across all demographic groups. Thus, if least-cost optimization capacity expansion models remain the dominant decision-making paradigm, strict low-carbon or renewable energy technology mandates will have the greatest likelihood of achieving national distributional energy equality. Decarbonization is essential to achieving climate goals, but myopic decarbonization policies that ignore co-pollutants may leave Black and high-poverty communities up to 26–34% higher PM$_{2.5}$ exposure than national averages over the energy transition.

As countries push for electricity system decarbonization, there is a risk that electricity transition investment can lead to outcomes that worsen social inequalities if marginalized groups are excluded from the benefits due to explicit exclusion or implicit human biases[1–3]. Thus, there is large uncertainty regarding the degree to which decarbonization policies will exacerbate or alleviate social inequalities and how they will impact co-pollutants (nitrogen oxides (NO$_x$), sulfur dioxide (SO$_2$), and particulate matter (PM)) emissions stemming from the electricity sector. Currently, most national electricity planning models investigate how the nation can decarbonize the electricity sector using least-cost optimization[4], without considering how different decarbonization strategies impact the distribution of air pollution concentrations across sub-national regions[5–7].

Achieving distributional energy justice requires that there is an equitable distribution of a society's technological and environmental risks, harms, and benefits[8]. In our paper, the harms and risks stem from air pollution exposure, and the benefits relate to the reduction of air pollution exposure following power plant retirements. Incorporating key principles from distributional energy justice begin to address the gap in electricity planning models by identifying how the benefits and harms of future energy transitions will be shared across a nation[8,9]. Even if income groups do have equal air quality concentrations, there still lies a disproportionate burden on lower income communities because of historical disadvantages, and less access to healthcare facilities when compared to wealthier communities[10]. We add to the literature by evaluating the environmental sustainability (i.e., national

[1]Engineering and Public Policy, Carnegie Mellon University, Pittsburgh, PA, USA. [2]Civil and Environmental Engineering, Carnegie Mellon University, Pittsburgh, PA, USA. ✉e-mail: tgoforth@andrew.cmu.edu; dnock@andrew.cmu.edu

air pollution emissions) and equality (i.e., distribution of air pollution) across eight national decarbonization strategies focusing specifically on the electricity sector. Our equality analysis focuses on the distribution of air pollution emissions, with total equality defined as each community having equal air pollution emissions from the electricity sector. A key contribution of our work is highlighting how the benefits of decarbonization (i.e., reduced emissions) impact different demographic groups.

Four sustainability dimensions (economic, environmental, social, and technical) often are used to measure national and regional sustainability[11–13]. While much of the literature addresses environmental sustainability from a national perspective (i.e., total emissions across a country), there is a need for a deeper understanding of how different decarbonization pathways will affect pollution distribution across vulnerable groups (distributional equality)[14]. Based on the review of energy transition literature, Kohler et al. illuminate the need for exploration of how energy transitions may place an undue burden on regions with high poverty rates or low-income populations[15]. Likewise, in a review of energy justice literature, Carley et al. indicate that it is known that energy transitions may exacerbate inequalities[16]. However, these two reviews indicate that there is a gap in understanding the magnitude and geographic distribution of energy inequalities and how energy systems transitions will impact the four dimensions of sustainability[15,16].

At a national level, air pollution is responsible for 100,000 to 200,000 excess deaths every year in the US and severe health effects, like lung, heart, and brain diseases[17–19], and these effects are often greatest felt in minority communities[20–23]. As the nation decarbonizes the electricity sector, air pollution across the US is likely to improve, but the distribution of air pollution may not be equal throughout the energy transition (e.g., some regions may be left with higher concentrations of pollution and more negative health impacts). Some studies have investigated the air quality co-benefits of decarbonization policies (i.e., additional reductions in other emissions like $PM_{2.5}$). In general, mitigating greenhouse gas emissions results in positive co-benefits in $PM_{2.5}$ emissions[24–26] at the system level [24–26], but there is still uncertainty regarding the spatial distribution of these emission reduction benefits, especially within vulnerable communities.

Multiple studies have investigated the air pollution exposure disparities across different racial and income groups using retroactive analyses, finding that low-income, Black, Asian, and Hispanic communities were exposed to higher levels of $PM_{2.5}$ in the US in 2000, 2014, and 2016[27,28], which stem from historical policy inequities[29]. While it is valuable to understand the level of historical injustices, countries also need a framework that evaluates future disparities in air pollution distributions across the nation under different decarbonization plans in order to mitigate and reduce future inequalities. Here we create a forward-looking analysis framework for assessing how decarbonization benefits will be shared across different demographic groups by tying a capacity expansion model with a local equality analysis.

A host of proposed models can evaluate the sustainability of electricity system transitions[4], with the most prevalent being least-cost optimization models. In these models, system-level environmental sustainability metrics are calculated after the optimization has been solved or integrated as one of the constraints[14,30]. Social dimensions (i.e., equality, equity, and justice) are often nonexistent, used as assumptions in models, or analyzed separately from capacity expansion models[14,15,31]. Thus, economic optimization drives the model decision-making, while environmental and social factors are considered a constraint or post-analysis at the system or sometimes country level[32], in which low spatial resolutions may miss specific impacts on people or communities. Our analysis addresses these limitations by quantifying how national energy transition policies impact sub-national equality at a high spatial resolution, designed using a least-cost paradigm for power plant investment decisions.

While least-cost optimization models often exclude equality considerations, some papers have integrated equality and distributional analysis into the electricity system decision making paradigm. One paper investigated the social and environmental implications of expanding power systems in developing countries with little to no existing infrastructure[33] at a subnational level. In Nock et al., the primary goal was to investigate how different stakeholder preferences towards equality (i.e., distribution of electricity access) impacted power grid construction[33]. However, the authors did not investigate how the distribution of air pollution emissions would change under different decarbonization strategies or the distributional impacts at a high resolution. Sasse and Trutnevyte[32] investigated the sustainability and equity impacts of reaching electricity sector targets across European countries. While this paper highlights four different optimization objective scenarios (base case, cost, equality, and renewable generation), their focus is on intercountry equality considerations[32], which miss local level equality impacts.

Sergi et al. perform a forward-looking analysis to investigate the impact of including co-benefits of decarbonization by including damages from air pollution in the objective function, but they do not explore distributional energy justice or equality of air pollution across different decarbonization scenarios[34]. Dimanchev et al.[35] investigate the co-benefits of different decarbonization policies on the rust belt and the regional distribution of these co-benefits but also do not investigate the impacts on different demographics. Luo et al.[36] investigate the air pollution benefits across different demographic groups in Texas by internalizing health impacts in energy planning. Mayfield[37] investigates the air pollution impacts on mortalities based on future coal plant retirements under two scenarios (no policy change and net zero) and across different demographic groups using a multi-objective energy and equity model. Burtraw et al. use a capacity expansion model tied to a reduced complexity model to investigate the air pollution benefits of reaching climate goals by 2030 across counties and different demographic groups. Jordaan et al.[38] investigate the impact of countries using natural gas as a bridge fuel in energy transitions and highlight carbon mitigation opportunities, but do not investigate the equity implications. We build on this work by investigating how least-cost optimization (dominant decision paradigm) for energy planning in the US impacts local equality objectives across eight decarbonization scenarios, some of which include 80-100% renewable penetration, national carbon caps, or 100% low carbon technology requirements. From this analysis, we can gain policy insights on how different decarbonization scenarios will impact equality goals throughout the energy transition while also quantifying how different groups will be burdened with air pollution.

In this work, we investigate and quantify the distributional equality of air pollution impacts at a local scale by tying a capacity expansion model with a sustainability and equality analysis. The capacity expansion model optimizes the US electricity sector from 2010 to 2050 on a least cost paradigm[5]. The electricity generation from the capacity expansion model are fed into the environmental sustainability model, which calculates air pollution emissions at the regional and national levels. We investigate local air pollution concentrations and associated disparities using a reduced complexity air pollution model, which ties source emissions to their transport, deposition, and where pollutants end up as ambient concentrations. Using this model, we build a high-resolution analysis to quantify air pollution concentrations across different demographic groups at the census tract level. Specifically, our environmental equality analysis examines the way national decarbonization policies could impact the distribution of air pollution concentrations across different demographic groups (e.g., median income, poverty, and race or ethnicity). Our air pollution analysis focuses on the generation of co-pollutants ($NO_x$, $SO_2$, and $PM_{2.5}$). $CO_2$ emissions will impact global greenhouse gas emissions and climate change. $CO_2$ emissions and system-level co-

pollutant emissions are the focus of our national decarbonization analysis, while co-pollutant emissions and their local effects are the focus of our equality analysis.

## Results

### Decarbonization scenarios

We ran eight decarbonization scenarios for this analysis, summarized in Table 1. The base case (Scenario A) assumes all current carbon and energy policies remain in place, like state renewable portfolios, the Cross-State Air Pollution Rule, investment and production tax credits, and regional cap-and-trade policies. All other scenarios include these policies as well. The carbon cap scenarios were defined by estimations in $CO_2$ reductions from the Energy Innovation Technology & Policy LLC modeling to US Nationally Determined Contributions from the 2015 Paris Agreement (Scenario B) and emissions reductions required to keep global warming below 1.5 °C (Scenario C)[39]. The Energy Innovation Technology & Policy LLC's model that simulates decarbonization scenarios deploys a mix of different policies to reduce emissions from all sectors. We focus on the electricity sector and use their estimations in reductions in emissions to reach either US nationally determined contributions (NDC) or the 1.5 °C pathway. Scenario B, which implements a carbon cap based on US NDC, allows for an increase in carbon emissions from 2040 to 2050 due to the policy assumptions made by the Energy Innovation Technology and Policy LLC modeling tool.

The national technology mandate scenarios (Scenarios D-H) were defined by assuming renewable energy and low carbon deployment began at 20% in 2020 and linearly increased to the mandate year (either 2035 or 2050). The technologies included in the national renewable energy mandate scenarios (Scenario D: 80% RE by 2050, Scenario E: 100% RE by 2035, and Scenario F: 100% RE by 2050) are solar photovoltaic (PV), concentrated solar power (CSP), onshore and offshore wind, biopower, hydropower, geothermal, landfill gas, pumped hydropower storage and battery storage. The national low carbon mandates (Scenario G: Low Carbon by 2035 and Scenario H: Low Carbon by 2050) allow generation from the same renewable energy technologies specified in the national renewable energy mandates, as well as generation from natural gas carbon capture and storage (CCS) and nuclear power plants. For technology costs, all scenarios assume the mid-case of the 2019 Annual Technology Baseline from the National Renewable Energy Lab[5]. Numerical inputs of the carbon caps and national technology mandates can be found in SI Table S-1.

### National energy transitions under decarbonization goals

We investigate the environmental impacts (i.e., total air pollution emissions) and equality (i.e., regional distribution of air pollution) of different electricity generation investment strategies under eight decarbonization strategies over 40 years (shown in Table 1 above).

Figure 1 shows the annual generation by technology for the decarbonization scenarios. For Scenario A (the Base Case), which implements no additional carbon constraints or policies, coal, natural gas, and nuclear mainly supply generation throughout the 2010–2050 timespan. By 2050 we see coal generation decrease to 7.5% of total generation (0.41 petawatt-hours (PWh)), natural gas generation slightly increases to 20.0% (1.08 PWh), onshore wind generation increases to 33.8% (1.83 PWh), and solar PV generation increases to 20.9% (1.14 PWh) of total generation. The carbon cap scenarios (B and C), which place a strict limit on $CO_2$ emissions from the electricity sector, achieve their carbon caps primarily through deploying solar PV and onshore wind. In both scenarios, wind and solar represented <3% of the generation in 2010. Still, by 2050 we see solar PV and onshore wind generation supplying 20–30% or 37–50% of total generation in 2050, respectively. Scenario C specifically sees the complete retirement of coal by 2035 and almost complete retirement of natural gas by 2050 (0.2% of generation). However, contrasting to Scenario C, the carbon cap defined in Scenario B allows an increase in $CO_2$ emissions, resulting in an increase in coal generation from 2040 to 2050 (1.67% of generation in 2040 to 4.67% of generation in 2050).

Scenarios with an implemented national renewable energy mandate (D, E, and F) invest the majority in onshore wind generation to meet their renewable energy mandates. By 2050, onshore wind represents approximately 50% of generation in all three scenarios. Scenarios D, E, and F also see large solar deployment due to the implemented mandate. Scenario E deploys the highest generation of solar PV, CSP, biopower, and battery storage to meet the 100% renewable requirement by 2035, with solar PV technology representing 35.0% of total generation by 2040. Solar PV is still a significant contributor to generation in the other seven scenarios, with solar PV supplying at least 15–20% of US generation in 2040 in each scenario.

Natural gas in the low carbon scenarios (G and H) is relied on until their low carbon requirement year, when natural gas is retired due to the mandate and replaced with natural gas CCS. Thus, natural gas would likely continue to provide 10–20% of the total generation needs without a low carbon mandate. See Table S-4 in SI for a summary of generation by technology and scenario.

**Table 1 | Description of decarbonization scenarios and their implemented policies in ReEDS (see SI Table S-1 for a description of ReEDS carbon policy inputs for each scenario)**

| Scenario | Scenario description | Scenario approach | Source |
|---|---|---|---|
| A | Base | Includes all current policies and standards (state renewable portfolio standards, tax credits, etc.) but implements no new carbon policies | [5] |
| B | US NDC | United States Nationally Determined Contributions via the 2015 Paris Agreement. Carbon cap implemented in ReEDS to follow emissions allotted | [39] |
| C | 1.5 °C Pathway | Based on policy required to maintain global warming under 1.5 °C. Carbon cap implemented in ReEDS to follow emissions allotted | [39] |
| D | 80% RE 2050 | National renewable energy mandate implemented beginning in 2020 at 20% and increased linearly to 80% renewable energy in 2050 | [5] |
| E | 100% RE 2035 | National renewable energy mandate implemented beginning in 2020 at 20% national RE generation and increasing linearly to 100% RE generation in 2035 | [66] |
| F | 100% RE 2050 | National renewable energy mandate implemented beginning in 2020 at 20% national RE generation and increasing linearly to 100% RE generation in 2050 | [5] |
| G | Low Carbon 2035 | National technology mandate implemented beginning in 2020 at 20% and increased linearly to 100% renewable energy, natural gas CCS, and nuclear in 2035 | [67] |
| H | Low Carbon 2050 | National technology mandate implemented beginning in 2020 at 20% and increased linearly to 100% renewable energy, natural gas CCS, and nuclear in 2050 | [67] |

## National environmental sustainability

Figure 2 shows the impacts of national operating emissions ($CO_2$, $NO_x$, $SO_2$, and $PM_{2.5}$) from the changing power plant profiles of different decarbonization scenarios 2010 to 2050. We define operating emissions as emissions produced directly from the power plant creating electricity. From Fig. 2, we see all pollutants have similar trends, with emissions decreasing through 2050 but at varying magnitudes across scenarios. Scenario A (base case) is an upper bound for national emissions across all pollutants in our analysis, indicating that implementing carbon or technology mandate policies represent a benefit compared to the do-nothing case.

By 2035, operating emissions from Scenarios C, E, and G are under 100 megatonnes (Mt) $CO_2$ emissions. Coal has entirely retired by 2035 in these scenarios, so it does not contribute to emissions, and natural gas or natural gas CCS contributes under 10% of generation. Fossil fuels like coal and natural gas are the main drivers of $CO_2$ and co-pollutant emissions from the electricity sector (see the breakdown of emissions by technology in SI Figs. S-3–S-6). Therefore, by phasing out coal and natural gas plants, emissions from co-pollutants like $NO_x$, $SO_2$, and $PM_{2.5}$ also fall significantly, with $NO_x$ levels at or below 0.02 Mt, $SO_2$ levels below 0.003 Mt, and PM levels below 0.002 Mt. $PM_{2.5}$ emissions (d) in Scenario E rise from 2035 to 2050 due to investments in biopower to maintain the 100% renewable energy mandate. Because of this, Scenarios C and F have lower levels of $PM_{2.5}$ emissions in 2050.

We also investigate the national ratio of $PM_{2.5}$ emissions per megawatt-hour of annual national generation to obtain national

emissions rates across the decarbonization scenarios (SI Fig. S-7). We find that the 100% renewable energy scenarios have the lowest $PM_{2.5}$ emissions per national annual megawatt-hour generation by their mandate year. Interestingly, we see that in the absence of a strict renewable energy mandate, the 1.5 °C decarbonization pathway (Scenario C) often has the lowest or second lowest ratio over the entire modeling horizon from applying an aggressive carbon cap. This most likely stems from Scenario C (1.5 °C decarbonization pathway) retiring the entire coal fleet in the same time period as Scenario E (100% RE by 2035).

## Air pollution distribution

While national-level emissions analyses are important for measuring progress across the energy system as a whole, regional inequalities resulting from energy transitions can manifest in the unequal distribution of air pollution concentrations. The operational co-pollutants will broadly impact a community's health in nearby regions, so operating emissions reductions of these pollutants will result in regional health benefits[40–43]. Therefore, we present an analysis of operating co-pollutant emissions ($NO_x$, $SO_2$, $PM_{2.5}$) to illuminate how different decarbonization scenarios could impact air pollution distribution locally. $PM_{2.5}$ concentrations throughout the results include both primary $PM_{2.5}$ (directly from the power plant) and secondary $PM_{2.5}$ (formed from $NO_x$ and $SO_2$). $NO_x$ and $SO_2$ results are actual concentrations of $NO_x$ and $SO_2$, not secondary $PM_{2.5}$ formed by these pollutants.

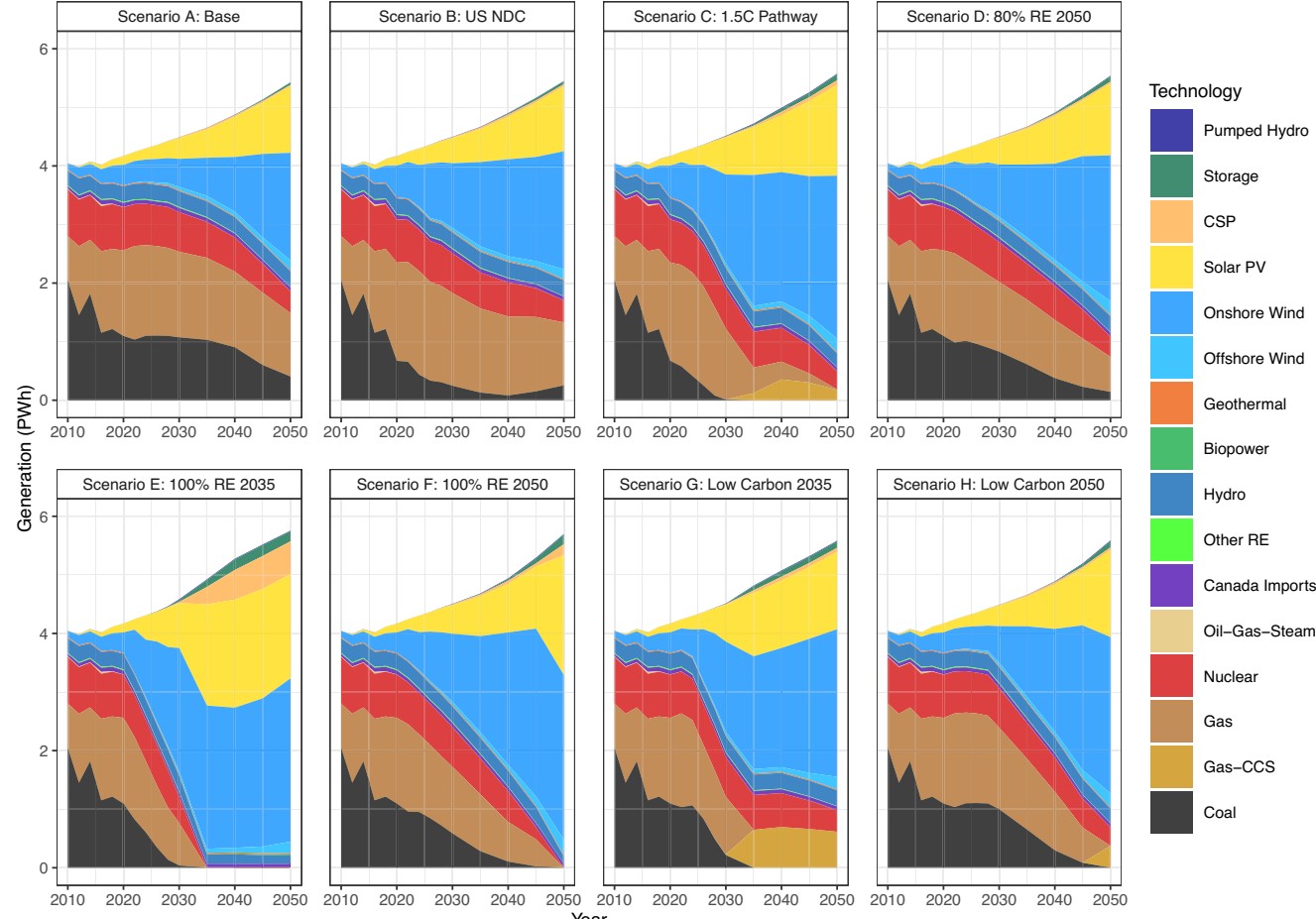

**Fig. 1 | Annual generation mix (PWh) 2010–2050 by technology for each decarbonization scenario resulting from the ReEDS model.** We highlight that the renewable and low carbon technology mandates accommodate additional energy needs primarily through expanded wind and solar generation investments. We see that the base case, US NDC, and 80% renewable energy decarbonization pathways retain coal generation through 2050.

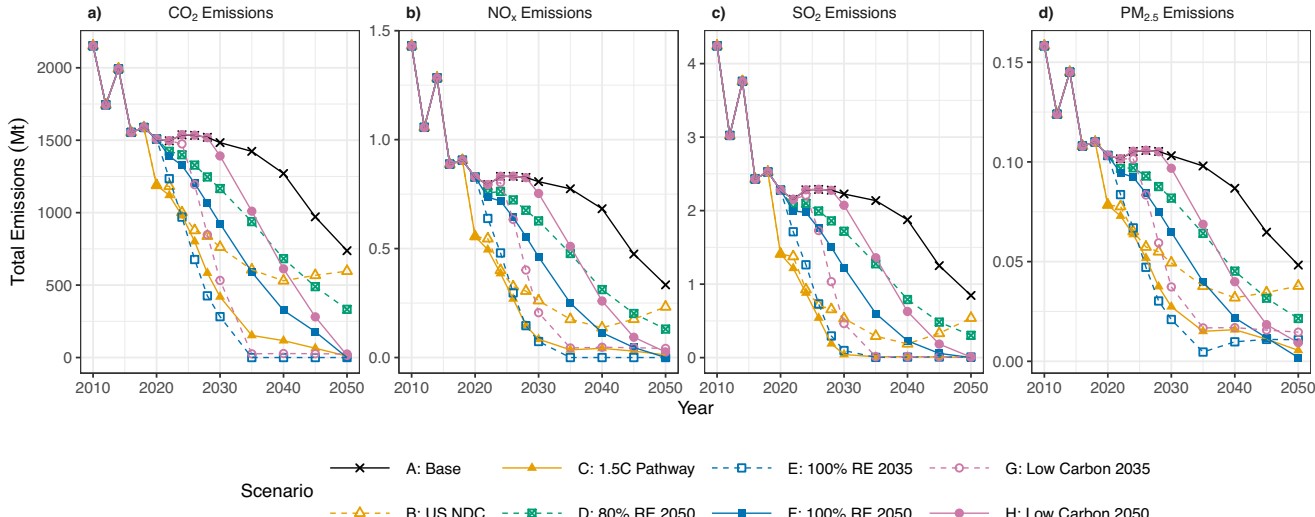

**Fig. 2 | National operating emissions across scenarios 2010–2050 (megatonnes, Mt).** The emissions shown here are: **a** $CO_2$ operating emissions, **b** $NO_x$ operating emissions, **c** $SO_2$ operating emissions, and **d** $PM_{2.5}$ operating emissions. Note that the y-axes are not consistent. We see the base case (black line) as an upper bound on all emissions types and Scenarios C (solid yellow line) and E (dotted blue line) as a lower bound across all emissions types. Scenario E emissions reach close to zero by 2035, its mandate year, and remains close to zero by 2035–2050 for $CO_2$, $NO_x$, and $SO_2$ emissions. However, $PM_{2.5}$ emissions in Scenario E rise because of investments in biopower, which help maintain the 100% renewable grid but still have co-pollutants that will be emitted.

Once emitted from the power plant, air pollution travels through the atmosphere affecting pollutant concentrations and causing health impacts[44]. We use a reduced complexity air pollution model, the Intervention Model for Air Pollution (InMAP), to simulate air pollution travel and understand where emissions are becoming ambient concentrations after being emitted from power plants. Figure 3 displays the annual average total $PM_{2.5}$ concentrations across census tracts for 2020, 2035, and 2050. The $NO_x$ and $SO_2$ concentration distribution reported similar trends to $PM_{2.5}$ (see SI Figs. S-8 and S-9). $PM_{2.5}$ concentrations in 2020 across scenarios have exposures over 1.0 µg/m³ in the Midwest and Eastern US. In 2035, Scenarios A, D, and H have concentrations over 1.0 µg/m³ located in the Eastern US and from Ohio to Iowa. Meanwhile, $PM_{2.5}$ concentrations in Scenarios C, E, and G are under 0.25 µg/m³ across all regions by 2035 because of an aggressive carbon cap (C) or clean technology mandates (E and G). Similarly, when Scenarios F and H reach their 2050 mandate year, $PM_{2.5}$ exposure is under 0.25 µg/m³ across all regions, indicating that total equality of air pollution concentrations is achieved when the technology mandate year is met (2035 or 2050). As a benchmark, the US Environmental Protection Agency's National Ambient Air Quality Standard for annual average secondary $PM_{2.5}$ concentration is 15 µg/m³ [3 45]. While the levels in our simulation (around 1 µg/m³) are lower than this standard, we note that this only accounts for air pollution from electric generating units. Thus, care must be taken to account for air pollution emission changes in other sectors.

**Air pollution concentrations across vulnerable groups**

Beyond regional analyses that measure the magnitude of air pollution, it is useful to understand the distribution of operating emissions across different demographic and socioeconomic indicators (race, ethnicity, income, poverty, etc.) across regions. This investigation shows the impact of different energy transitions on vulnerable regions. We focus on operating $NO_x$, $SO_2$, and $PM_{2.5}$ emissions, due to the local health impacts of these co-pollutants.

Figure 4 compares the average annual population weighted concentration of $PM_{2.5}$ across different race or ethnicity groups. We see that Black communities are exposed to higher concentrations of $PM_{2.5}$ over the energy transition until a renewable or low carbon technology mandate is >80% penetration or a carbon cap requires <400 Mt $CO_2$

emissions nationally. Based on these results, Black communities are at risk for higher $PM_{2.5}$ concentrations and its associated health impacts in energy transitions but also see the largest absolute reductions in air pollution exposure (see SI Fig. S-16). Without any decarbonization policies (the base case), Black communities are exposed to up to 34% more air pollution compared to the national average between 2020–2050. The percent change of $PM_{2.5}$ concentration across race and ethnicity changes at the same rate within each scenario (see SI Fig. S-17), indicating that the starting point of air pollution impacts future exposure throughout the energy transition. The group with the second highest $PM_{2.5}$ concentrations are non-Hispanic White populations. $NO_x$ concentrations showed similar trends, with Black communities exposed to higher concentrations of $NO_x$ (SI Fig. S-10). $SO_2$ concentrations show that non-Hispanic White and Black communities are equally exposed to the highest concentrations of $SO_2$ (SI Fig. S-11). When investigating the regional distribution of race and ethnicity groups across the US (SI Fig. S-20), we find that regions with census tracts over 95% non-Hispanic White and census tracts that are >10% Black have some of the highest $PM_{2.5}$ concentrations.

We find the 1.5 °C pathway (Scenario C) and the 100% renewable energy deployment by 2035 pathway (Scenario E) shirks the inequality between racial groups the fastest (by 2030). This is important for recognitional justice because disadvantaged communities start out with severe excess burden compared to their counterparts[28]. Scenario E has the highest absolute reductions in $PM_{2.5}$ concentrations across racial groups (SI Fig. S-16). By 2050, the decarbonization pathway with the least improvement in $PM_{2.5}$ concentrations for Black communities is the US NDC carbon cap pathway (Scenario B). In 2050 we see Black communities having a concentration of 0.25 µg/m³, compared to the Base Case (Scenario A) where Black communities experience a $PM_{2.5}$ concentration of 0.38 µg/m³. Thus, choosing any decarbonization pathway will achieve at least a 33% improvement over the base case in 2050.

We also investigate the change in $PM_{2.5}$ with respect to the change in $CO_2$ in 2050 across race and ethnicity groups (see SI Table S-6). If the ratio is less than one, it indicates that $CO_2$ is reduced faster than $PM_{2.5}$. We can use this ratio to understand the rate of $PM_{2.5}$ reductions across different decarbonization pathways and racial groups. However, the ratio does not capture the magnitude of reductions, which could

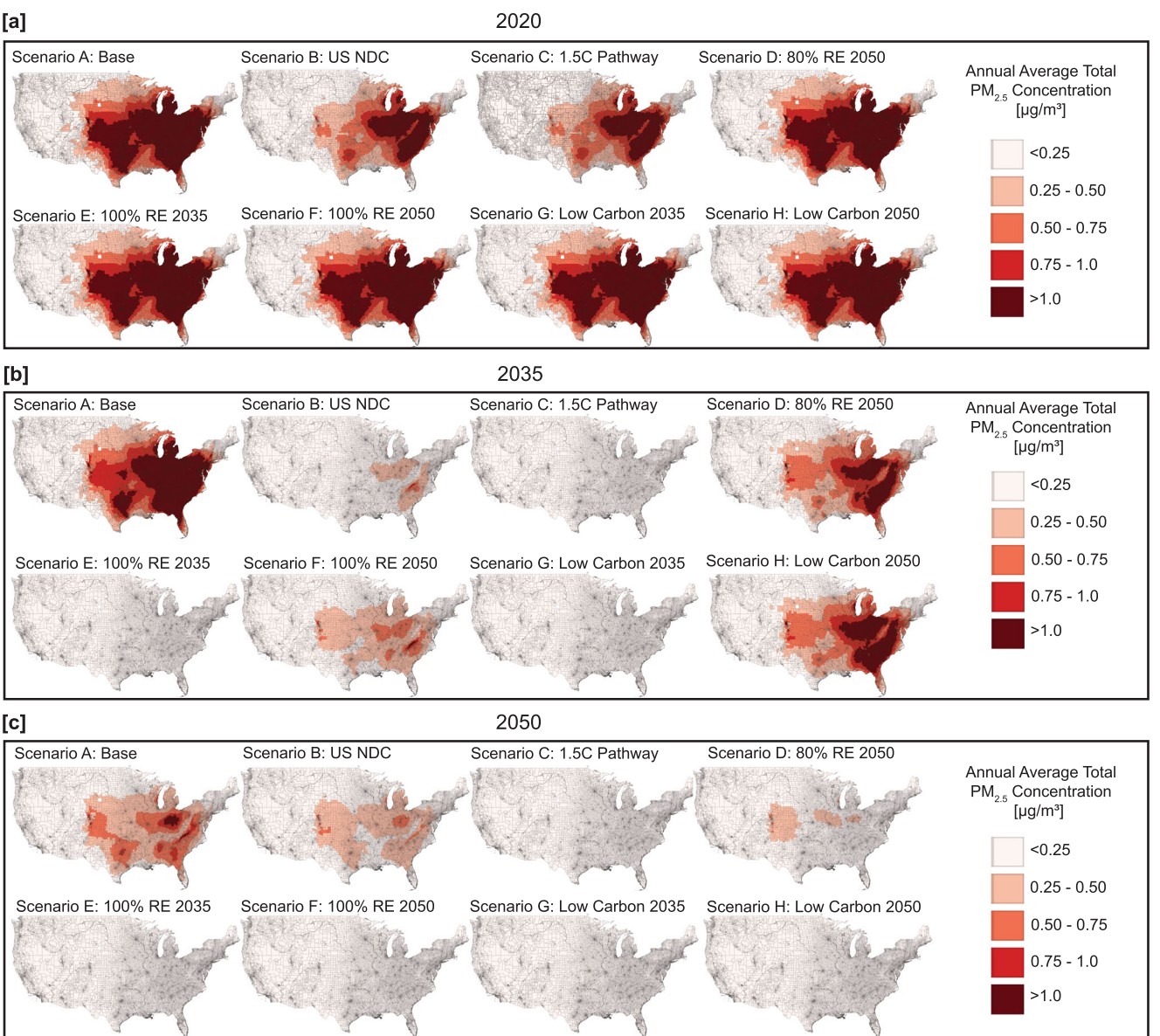

**Fig. 3 | Regional distribution of PM$_{2.5}$ across decarbonization scenarios for 2020, 2035, and 2050.** Total PM2.5 air pollution from power plants across scenarios: **a** 2020, **b** 2035, and **c** 2050. By 2035, the only scenarios with 100% of regions below the threshold of 0.25 μg/m³ include the aggressive carbon cap (Scenario C) and the ones with a technology mandate with a 2035 goal of 100% low carbon or renewable technologies (Scenarios E and G). The distribution of NO$_x$ and SO$_2$ for 2020, 2035, and 2050 can be found in SI Figs. S-8 and S-9.

impact decision-making. We see that Scenario D (80% RE by 2050) has the highest ratio of reductions in PM$_{2.5}$ concentration reductions per national reductions in CO$_2$ for all race and ethnicity groups. This indicates that while Scenario D does not reach zero CO$_2$ or PM$_{2.5}$ emissions, it reduces local pollutants at a faster rate per CO$_2$ reduction than other scenarios.

Figure 4 also shows that Hispanic, Asian, and Indigenous communities in the US are exposed to less PM$_{2.5}$ from power plants in the electricity sector over the course of the energy transition. This is consistent with historical trends where Black and non-Hispanic White communities have been disproportionately impacted by coal-fueled power plants[27,28]. Since we are only investigating emissions and air pollution concentrations from the electricity sector, our analysis does not capture air pollution injustices caused by transportation, industrial, or residential cooking activities. Research has shown that air pollution from these activities is where Hispanic and Asian communities are disproportionately affected[19]. Li et al. show that Indigenous

communities are also disproportionately burdened by PM$_{2.5}$ concentrations when accounting for all emission sources[28,44].

We present the population weighted PM$_{2.5}$ concentrations across census tracts by poverty rate in Fig. 5. Air pollution concentrations in census tracts with poverty rates greater than 70% are higher than all other census tracts before technology mandates that require less than 80% renewable energy or 100% low-carbon technologies. Census tracts with greater than 70% poverty have the highest concentrations of PM$_{2.5}$ over the energy transition until more than 80% of renewable energy is deployed. Therefore, reductions in air pollution benefit the highest-poverty census tracts the most since they are burdened with higher exposures. We also note that cumulative exposure combined with community-level stressors will impact the health disparities among high poverty rate groups[46]. By 2050, the decarbonization scenario with the least improvement in PM$_{2.5}$ for census tracts with poverty rates above 70% is Scenario B (US NDC carbon cap) with PM$_{2.5}$ concentrations at 0.25 μg/m³. Further, without implementing any

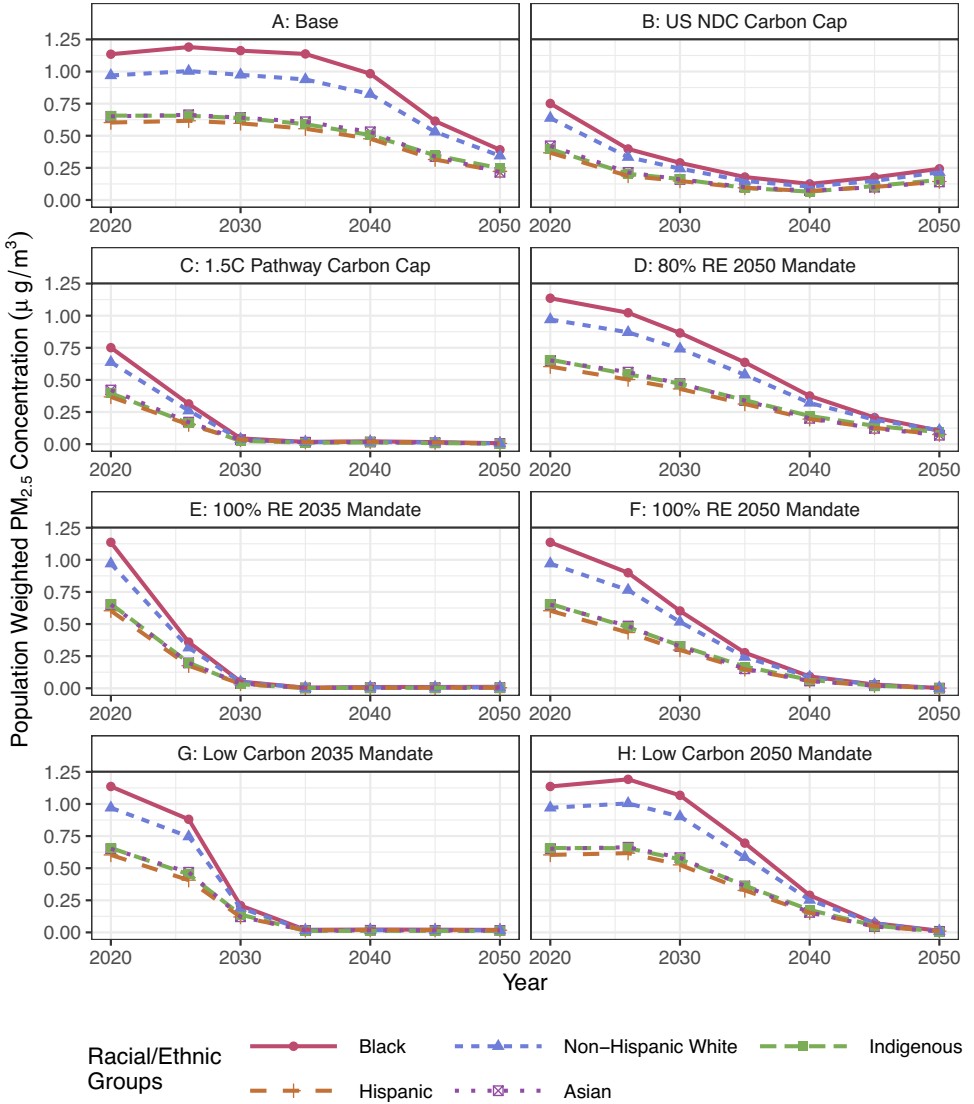

Racial/Ethnic Groups

Black — Non–Hispanic White — Indigenous — Hispanic ··· Asian

**Fig. 4 | Population weighted average annual PM$_{2.5}$ concentrations (in µg/m³) across different race and ethnicity groups for each scenario from 2020 to 2050.** We see that Black communities in the US are exposed to higher concentrations of PM$_{2.5}$ in 2020, which is consistent with historical impacts[28]. Over the energy transition, Black communities are exposed to higher concentrations of PM$_{2.5}$ until a technology mandate is >80% renewable energy (Scenario D in 2050, Scenario E in 2035, or Scenario F in 2050), 100% low carbon energy (Scenario G in 2035 and Scenario H in 2050), or carbon cap that requires emissions of CO$_2$ <400 Mt (Scenario C in 2030).

decarbonization policies (Scenario A, the base case), communities with poverty rates >70% see a maximum of 26% higher PM$_{2.5}$ concentrations over the energy transition.

Figure 6 shows the population weighted annual average PM$_{2.5}$ concentration across the highest (>$150k), lowest (<$25k), and mid ($100k–$125k) income groups. While we see that the lowest income group has the highest PM$_{2.5}$ exposure, the difference between the three groups is <0.20 µg/m³ PM$_{2.5}$ concentration across all scenarios and years. However, higher-income people have more access to health care facilities and insurance, so low-income groups with the same concentrations may be left worse off since they cannot access healthcare for health impacts from air pollution as easily[10].

Overall, we find that median income is less of an indicator of population weighted air pollution concentration than race, ethnicity, or poverty within a region. Within each median income group, Black communities are exposed to higher concentrations of population weighted air pollution than any other race or ethnicity group (see SI Fig. S-15).

PM$_{2.5}$ and NO$_x$ population weighted concentrations across income groups do not see significant differences (less than 0.2 µg/m³ before 2035 in Scenario A: Base Case), but we find that there is a larger disparity between the highest and lowest income group in population weighted SO$_2$ concentrations (up to 0.3 µg/m³ before 2040 in Scenario A: Base Case) (SI Fig. S-13). This disparity is likely due to coal plant placement in low-income areas[47]. Coal power plants are the main contributor to SO$_2$ emissions, as seen in SI Fig. S-5.

## Discussion

We investigated how national level decarbonization policies translate to national emissions and the distribution of air pollution at the local level. Our analysis finds that no decarbonization scenario reaches operating emission distributional equality until they meet their mandate year of 2035 or 2050 (Scenarios C, E, F, G, and H). However, there are clear trade-offs between national emissions reductions and distribution of air pollution across regions: to maintain a 100% renewable requirement after 2035, biomass power plants are deployed, which emit SO$_2$ and PM$_{2.5}$. While biomass can be considered a carbon neutral

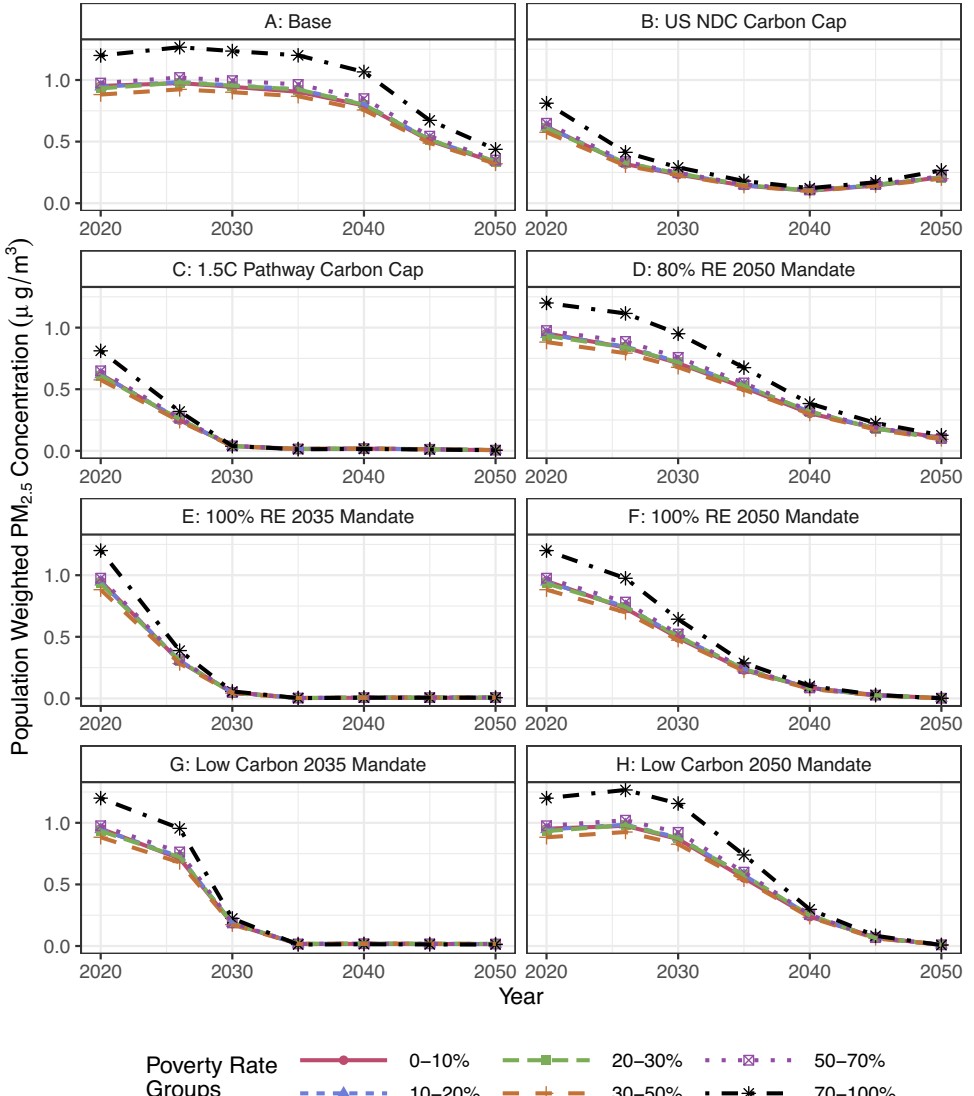

**Fig. 5 | Population weighted average annual PM₂.₅ concentration across poverty rates within census tracts for each scenario 2020–2050.** We see that census tracts with poverty rates >70% are burdened with the highest concentrations of PM₂.₅ across all scenarios and timelines. Figures S-18 and S-19 show the NOₓ and SO₂ concentrations across poverty rate groups, respectively.

source of energy, a policy question would be investigating the differences in where the emissions are emitted and absorbed. The emissions directly from biomass plants will negatively affect surrounding communities and cause greater inequality across distributional air pollution. We find that the carbon cap scenario, which aims to keep warming under 1.5 °C, has fewer national reductions in emissions compared to 2035 technology mandates but results in an equal distribution of air pollution (<0.25 μg/m³ concentrations) by 2050 for all co-pollutants. The 100% renewables by 2050 (Scenario F) and low carbon technology mandates (Scenario G and H) also see this trend. We note that there is considerable uncertainty in how emission regulation decisions will impact the achievability of high renewable deployment[48]. This further highlights the multiple objectives, factors, and often conflicting nature of energy transition planning.

When addressing the multi-faceted lens of decarbonization, it is important to weigh both the aggregate national emissions reductions and the distribution of those emissions reductions. For example, reaching 100% renewable energy by 2050 will help the US decarbonize its electricity sector entirely. However, before the mandate is reached in 2050, there are air pollution inequalities as the nation decarbonizes, with high-poverty and Black communities seeing the highest co-

pollutant concentrations. This result may be a byproduct of the least cost paradigm being the primary objective guiding technology deployment as well as historical trends. We also see that trends for who is exposed to higher concentrations of pollutants follow historical trends, so the distribution of air pollution over the energy transition using a least cost paradigm is dependent on the historical and current concentrations across demographic groups. Future work could investigate how historic inequities impact future air pollution distribution. This paper focuses on distributional justice (distribution of decarbonization benefits and co-pollutant burdens), and a subset of recognitional justice (impact on historically disadvantaged communities).

In this analysis, we created a framework for social impact assessments to identify who may be burdened with higher levels of air pollution in national decarbonization transitions. This could help advise decision makers on who loses in energy transitions and help build a policy to compensate them. The continued air pollution inequality from historical trends will exacerbate health impacts among the most vulnerable communities. Four scenarios reach zero (or close to zero) operating emissions by their mandate in either 2035 (E and G) or 2050 (F and H), but not beforehand. This result indicates that achieving

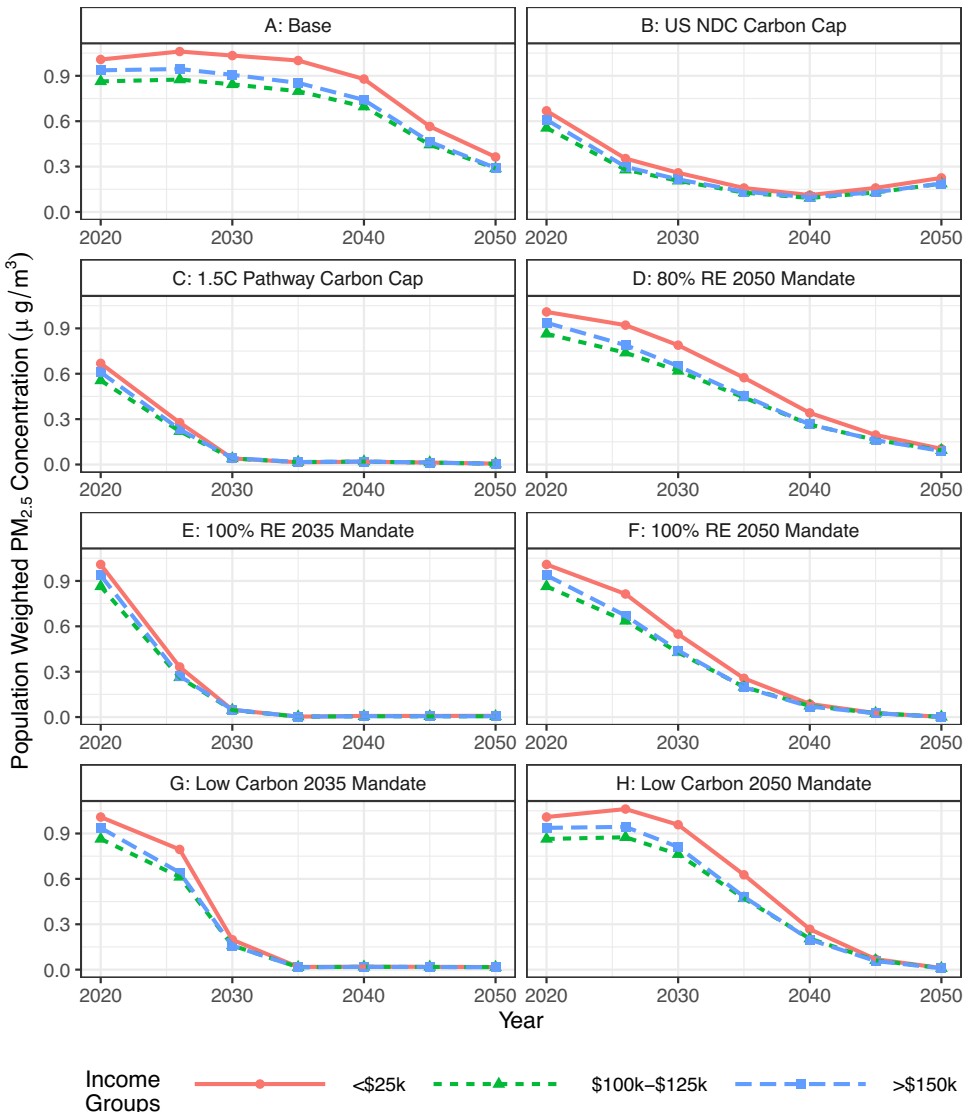

**Fig. 6 | Population weighted annual average PM$_{2.5}$ across each scenario 2020 to 2050 for the highest (>$150k), mid ($100k-$125k), and lowest (<$25k) income groups in µg/m³.** The other income groups fall in between the highest and lowest bounds. Results from all scenarios can be found in SI Fig. S-12, and NO$_x$ and SO$_2$ across income groups in Figs. S-13 and S-14, respectively.

deployment of more than 80% renewable energy or low carbon by the given mandate year can ensure an equal future beyond those years. However, beforehand, high-poverty and Black communities are burdened with the most air pollution. Further, while air pollution concentration equality may be reached at an 80% penetration of renewable energy, there are still fossil fuels in the mix in this scenario (10% natural gas generation and 2% coal generation in 2050). Continued emissions of CO$_2$ and co-pollutants from these power plants could negatively impact communities.

All scenarios with carbon policies implemented see improvements from Scenario A, which implements no additional carbon policies after 2020. There is at least a 20% reduction in co-pollutants concentrations in the lowest income group in 2050 in the other scenarios compared to Scenario A. However, a gap persists between the best-off and worse-off regions across all demographic variables and time periods we consider. If an equitable energy transition is the goal (i.e., one that reaches total equality), decarbonization policies in the absence of strict technology mandates and those guided by least-cost optimization capacity expansion models may fall short of environmental justice and equality goals. Thus, decisions regarding national

decarbonization pathways must have strict mandates for equality outcomes or be driven by an equality-focused paradigm. Two opportunities for future analysis present themselves. The first is to investigate how changing the decision-making paradigm (i.e., changing the optimization objective function) influences the equality outcomes between regions. The second is to investigate the trade-offs of air pollution distribution with other equality (e.g., distribution of costs and electricity bill increases), equity (e.g., health impacts), environmental (e.g., water consumption and land-use), and cost objectives. A deeper analysis of health impacts from energy transitions could also necessitate greater quantification of the monetary damages or deaths from air pollution[49–52].

Equitable energy transitions exist at the intersection of technical, economic, and social justice objectives[8,16,53–56]. Achieving the goal of an equitable energy transition requires a multi-disciplinary lens to understand who wins and loses in energy transitions. Our work begins to do this by using a least-cost optimization model coupled with a sustainability and equality analysis that measures air pollution across regions and demographic groups. This research is a first step in investigating an equitable energy transition by analyzing how national

**Table 2 | Operating emission rates used in environmental sustainability analysis [in g/kWh]**

| | CO₂ [g/kWh] | NOₓ [g/kWh] | SO₂ [g/kWh] | PM₂.₅ᵃ [g/kWh] |
|---|---|---|---|---|
| Biopower | 0 | 0 | 0.490 | 0.620 |
| Solar photovoltaic (PV) | 0 | 0 | 0 | – |
| Concentrated solar power (CSP) | 0 | 0 | 0 | – |
| Onshore wind | 0 | 0 | 0 | – |
| Offshore wind | 0 | 0 | 0 | – |
| Nuclear | 0 | 0 | 0 | – |
| Natural gas combustion turbine (CT) | 496 | 0.637 | 0.064 | 0.028 |
| Natural gas combined cycle (CC) | 337 | 0.058 | 0.015 | 0.019 |
| Natural gas CCS | 39.8 | 0.069 | 0.017 | 0.022 |
| Hydropower | 0 | 0 | 0 | 0 |
| Geothermal | 0 | 0 | 0 | 0 |
| Oil–gas–steam | 662 | 0.832 | 1.44 | 0.081 |
| Coal | 923 | 0.672 | 2.06 | 0.069ᵇ |
| Coal IGCC | 756 | 0.305 | 0.199 | 0.056ᵇ |
| Coal CCS | 97.0 | 0.392 | 0.256 | 0.072ᵇ |
| Cofire (biopower and coal) | 821 | 0.598 | 1.89 | 0.131 |
| Battery storage | 0 | 0 | 0 | 0 |
| Pumped hydropower | 0 | 0 | 0 | 0 |

See Table S-2 in SI for sources: dashed line indicates no reported value.
ᵃFor renewable energy and nuclear technologies, we assumed PM operating emissions were negligible.
ᵇAssumed Bituminous coal.

policies translate to subnational equality. We have shown that a single objective of minimizing cost leaves vulnerable groups at risk of existing in regions with higher air pollution concentrations throughout the transition. When crafting public policy for energy transitions, decision-makers can use this work as a source for indicating the need for holistic multiple objective approaches to energy system planning if we are going to ensure an equitable and sustainable future.

## Methods

Here we discuss the electricity system modeling, decarbonization scenarios, and sustainability and equality analyses. We conclude this section by discussing the limitations of our analysis. Our work investigates the air pollution equality of decarbonization scenarios at the census tract level in the US. We do this by tying a national capacity expansion model with an air pollution assessment and distributional equality analysis.

### Electric power system model

Our electricity system analysis uses the Regional Energy Deployment System (ReEDS) from the National Renewable Energy Lab (NREL) to define the resulting electricity generation profiles under different decarbonization scenarios, such as a carbon cap or national renewable portfolio standard, to reach different energy transition goals. The model outputs generator capacity, generator generation, system cost, generator retirements, and transmission data for regions in the US defined by ReEDS for each model year. We use these data to analyze the impact of air pollution equality of different decarbonization scenarios. ReEDS is a least-cost, linear program that minimizes the cost of the future electricity system subject to load, operating, and transmission constraints. ReEDS runs sequentially, meaning that each model year is solved individually before continuing to the next model year. Because ReEDS solves sequentially, the model has limited foresight

into model input changes over time; thus, it does not account for changes to policies or the market[5]. See Fig. S-1 for the spatial resolution of the model.

ReEDS implements a carbon cap or technology mandate as an exogenous input to the model. The carbon cap specifies allowed emissions in the US electric sector for each model year (2010 to 2050). The operating emissions generated from the system cannot surpass the specified yearly carbon cap in the model. The model will not continue to the next solve year until it can find a solution that meets the carbon emissions cap. The technology mandate specifies a percentage of chosen technologies (i.e., solar and wind) that are required generate a certain percentage of electricity each year. As with the carbon cap, the model must satisfy the share of generation from the specified technologies before continuing to the following year.

### Environmental sustainability (emissions) assessments

This analysis uses operating emissions to represent environmental sustainability. Operating emissions are classified by the emissions produced while the power plant is generating electricity.

Table 2 displays the emission rates used in this analysis. The national and regional emissions are calculated by multiplying the generation ($g$) for each technology ($n$) and each model year ($t$) by the emissions rate (in g/kWh) for each technology ($e_n$). See Table S-3 for heat rates and fuel emissions rates for each power plant. The emissions used in this analysis are carbon dioxide ($CO_2$), nitrous oxides ($NO_x$), sulfur dioxide ($SO_2$), and particulate matter with a diameter less than 2.5 microns ($PM_{2.5}$).

$$E = \sum_{i=1}^{n} g_{n,t}\, e_n \qquad (1)$$

We obtained operating emission rates for power plants from literature (See SI Table S-2 for sources). We assumed that operating emissions from renewable and nuclear sources were zero. For more information on emissions rates see the following sources: US Environmental Protection Agency (2020)[57] and the National Renewable Energy Lab (2019)[5].

### Air pollution equality assessments

The demographic metrics we use to evaluate equality in this analysis are median income, percent of the population in poverty, and race or ethnicity. We obtained these datasets from the American Community Survey (ACS) from the US Census Bureau[58] at the census tract level. We split regions into groups based on each metric better to understand the distribution of emissions across these equality metrics. For example, we grouped median income across census tracts at intervals of $25,000. Table S-5 in SI shows group intervals and their respective sample sizes. The race and ethnicity groups were defined by the ACS survey, which has population counts of each race and ethnicity group in each census tract.

We compare these equality metrics to air pollution concentrations to investigate the inequalities across regions for each decarbonization scenario. The air pollution model (InMAP) is tied to census tracts by taking the average concentration across modeling regions for each census tract, as detailed in Fig. 6. These estimations are from 2018, so they may not reflect what median income, percent in poverty, or demographics may look like in future time periods. Thus, one limitation is the lack of projection regarding human migration patterns at the subnational level, which may be impacted by rising temperatures and changing weather patterns.

### Air pollution across vulnerable groups

Within the US electricity sector, Black and non-Hispanic White communities have the most deaths per 100,000 people from $PM_{2.5}$ from

electric generating units[28]. To understand how energy transitions may exacerbate or reduce local air pollution disparities, we use a reduced complexity model, InMAP, to quantify where co-pollutants ($NO_x$, $SO_2$, and $PM_{2.5}$) end up as ambient concentrations after being emitted from power plants. Reduced complexity models are commonly used to evaluate the impact of air pollution disparities across race, ethnicity, and income groups[28,59] as well as the health impacts, estimated deaths, and monetary damages from air pollution in the US[18,49]. Figure 6 summarizes how the analysis estimates air pollution from the ReEDS region level to the census tract level. InMAP uses the quantities of power plant emissions (in kg) as shapefile inputs and uses area-weighting to allocate emissions from our electricity capacity expansion model (ReEDS) regions to the more spatially granular grid. Area-weighting means the model equally distributes emissions from the ReEDS regions to the InMAP grid based on the size of the InMAP region (see SI Eq. S-1 for the area-weighting equation). Then, InMAP calculates annual average emissions for $NO_x$, $SO_2$, and $PM_{2.5}$ using a reaction–advection–diffusion equation[52]. InMAP was tied to census tracts by taking the average concentration across modeling regions for each census tract, as detailed in Fig. 7 below.

InMAP calculates the annual average concentration of pollutants by using equations that account for the travel and deposition of air pollution (advection, mixing, chemical reactions, and deposition equations) across a variable spatial grid. Advection is the transport of air pollution from the wind. Mixing equations account for the transport of air pollution from turbulent mixing (not wind). Chemical reactions account for secondary $PM_{2.5}$ that $NO_x$ or $SO_2$ can form. Deposition equation estimates where and how much air pollution is deposited in the region. InMAP uses a variable grid with square grids ranging from 1 to 48 km as its spatial resolution[52]. See SI section C for a more detailed explanation of InMAP.

Air pollution concentration results are represented in population weighted air pollution concentrations (Eq. 2). Population weighted concentrations are calculated by summing the product of the population of a demographic group within a census tract ($p$) and the air pollution concentration ($c$). This is then divided by the total population ($P$) of a demographic group.

$$\frac{\sum (p * c)}{P} \qquad (2)$$

## Limitations and caveats

Our work presents a subnational analysis of national decarbonization strategies' environmental sustainability and equality impacts. Here we present some limitations and caveats for the work presented here.

The population and equality metrics data collected were in 2010 (population) and 2018 (equality metrics). Our calculations will likely change if we use a model to simulate population migration patterns across our modeling time horizon (2010–2050). However, there is evidence that low-income groups have fewer resources and thus are less likely to move over time[60], so our current metrics will likely have little change over time and are valid estimations and assumptions.

The ReEDS model operates over a 40-year time horizon, so there is inherent uncertainty in its outputs (i.e., location of power plants, generation of those power plants in each model year). See SI Section A for model assumptions and specifications made in this analysis. The ReEDS documentation by NREL documents all assumptions in model inputs and model construction[5]. One aspect of uncertainty is the temporal and spatial resolution of the model[61]. Due to a least-cost paradigm directing the model, its final outputs may vary as costs of technology fall, demand changes, or implementation of different policies. These cost decreases will vary by location, workforce, labor costs, and scarcity or abundance of input materials over time. The goal of our analysis was not to perfectly simulate which technologies would

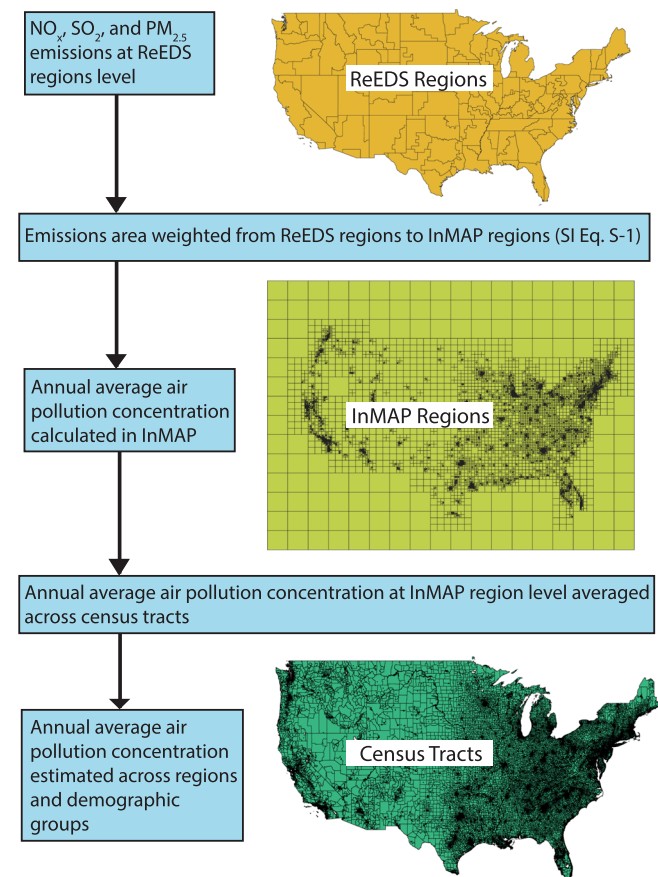

**Fig. 7 | Overview of equality analysis methods using a capacity expansion model (ReEDS) and a reduced complexity air pollution transport model (InMAP) to investigate where emissions are settling after being emitted from power plants.** Regions are downscaled from ReEDS regions to census tract level, and annual average air pollution concentrations are estimated from emissions at the ReEDS region level. Figure S-2 shows InMAP emission inputs. The shapefiles for the maps produced in this image are sourced from NREL ReEDS[5], InMAP output file[52], and ArcGIS[64].

be used in future generation mixes but to highlight how different decarbonization pathways might impact vulnerable groups.

Reduced complexity models, like InMAP, were created to offer an extensive air pollution model that does not require the expertise or computing power that chemical transport models need[52]. In a comparative analysis between InMAP and a chemical transport model, InMAP reported $R^2 = 0.90$ and a mean fractional bias of −17%[52], indicating InMAP results are within the bounds for its air quality results to be valid. However, there is also uncertainty in using a reduced complexity model. One point of uncertainty is the area-weighting of the emissions from the ReEDS region level to the InMAP grid level (see SI Eq. S-1 for area-weighting calculation). Emissions are distributed across the region and not at the power plant level because ReEDS does not identify power plant locations in its simulation. Future work could estimate ReEDS level emissions to the power plant level by allocating ReEDS level emissions to fossil fuel power plants.

While there are many possible impacts from decarbonization efforts in countries, we note that our analysis only investigates the air pollution impacts within the electricity sector in a developed country. Investigating decarbonization impacts from other sectors, like transportation, residential housing, and industry is an important next step in the understanding the full extent of inequities in the energy system and how they will be impacted by the energy transition. Deployment of

low carbon technology will also have different implications in emerging countries who wish to expand their electricity systems[62,63].

## Reporting summary
Further information on research design is available in the Nature Portfolio Reporting Summary linked to this article.

## Data availability
The models used in this analysis (ReEDS and InMAP) are both open-source tools. ReEDS requires R 3.4.4, Python 3.6.5, and GAMS 30.3. Python 3.8.3, ArcGIS Pro 3.0.2, QGIS 3.16.11, and R 3.6.2 were used to process data and create figures. The processed ReEDS generation outputs, regional emissions, InMAP air pollution raw outputs, InMAP air pollution processed data, and population weighted air pollution outputs data generated in this study have been deposited in this GitHub repository. The ReEDS decarbonization scenarios inputs are provided in the Supplementary Information file Section A. The census data used in this study are available from ArcGIS[64], and income and poverty data used in this study are available from ArcGIS[65]. The datasets of demographic data are shapefiles adapted from the 2010-2014 and 2011-2015 American Community Survey from the US Census Bureau, respectively.

## Code availability
The code used to perform the data analysis is available upon request.

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

## Acknowledgements

We would like to acknowledge that this work has been supported by National Science Foundation Grant 2017789. D.N. also acknowledges support from the Google Award for Inclusion Research and the Scott Institute for Energy Innovation, where she is an energy fellow. We would also like to thank the SPICE Lab at Carnegie Mellon University. Specifically, we would like to acknowledge the help of the undergraduate and Master's researchers who aided in the research process: Purva Bommireddy, Katrina D'Arms, Katherine Hart, Skylar McAuliffe, and Erin Percevault.

## Author contributions

T.G. designed and performed the analysis, outlined the analysis steps, lead the methodology development and paper writing, and edited the paper. D.N. formed the research idea, outlined the analysis steps, oversaw the research process, and edited and reviewed the paper.

## Competing interests

The authors declare no competing interests.
