## [Peer Review File · Nature Communications]

Air pollution disparities and equality assessments of US national decarbonization strategiesREVIEWER COMMENTS

Reviewer #1 (Remarks to the Author):

This paper examines how decarbonization pathways driven by a least-cost paradigm will impact air pollution inequality across different vulnerable groups in the United States. A key finding is that Black or poor communities have higher co-pollutant concentrations during the energy transition, until a national mandate requires full deployment of renewable or low-carbon technologies. The questions posed by the authors are important and policy-relevant. However, several aspects (including the method to obtain the distribution of air pollution at regional scale and across vulnerable groups) should be explained more carefully before the paper can be considered for publication. Some suggestions are offered below:

- In the Introduction:

- o You add to the literature by evaluating environmental sustainability and equality across eight national decarbonization pathways between 2010 and 2050. Is your paper the first one taking a forward-looking approach for this type of analysis? Or are there other papers that evaluate future distributional effects in connection to an energy systems model, and how does your analysis differ from theirs?

- o You only cite a couple of papers considering historic distributional effects, but in fact this literature is growing (see, among others, the recent work by Erin Mayfield and Ines Azevedo). Please add a more detailed review of the existing literature focusing on historic distributional effects.

- The paper would benefit from a better description of the scenarios. For example:

- o How do you obtain the annual emissions under the 2 carbon cap scenarios (B and C)?

- o What specific technologies/sources are included in the RPS targets in scenarios D-F?

- o The Low Carbon scenarios reach "100% renewable energy, natural gas CCS, or nuclear". What specific renewable energy technologies/sources are included? Are RE/gas+CCS/nuclear mutually exclusive, or can they be used jointly to reach the target?

- o Scenario B sees an increase in all emissions and coal generation from 2040 to 2050. Why?

- Figure 4: Looking at the solid red line (Black), can you explain why average PM2.5 concentrations rise until ~70%, and then decrease? Can you explain the trend in the dotted blue line (White)? Results for other minority groups (in particular, Latinx/Hispanic) seem to suggest that average concentrations of PM2.5 decrease as the share of Latinx/Hispanics increases – isn't this a counterintuitive results?

- Figure 5:

- o You write: "The points on the plot represent the minimum median income in each group (e.g., the point at \$0k represents the <\$25k income group)." Please clarify this statement. For example, what do the points at \$50k, \$100k and \$150k represent?

- o Why are some pollutants (e.g., NOx, PM2.5) increasing as median income increases?

- Please describe the InMAP model in more detail. You are currently referring to the model documentation for details, but the reader should be able to understand important features of the model used in your analysis without having to consult other papers.

- I would like to see a better description of the method to obtain the distribution of air pollution at regional scale and across vulnerable groups. This is a key part of your analysis, and should be discussed more carefully to help the reader understand the drivers of your results.

- Please describe how the ReEDS and InMAP models are linked in more detail.

- Some typos – please proofread carefully before submission.

Reviewer #2 (Remarks to the Author):

This article provides important information regarding equity implications of energy transition scenarios. This is an important topic, the methods used are scientifically rigorous, and the results support the claims that are made. The methods are well described.

I feel that the paper could be made even better by going a little further in the analysis of the data. For example if there were plots showing the absolute or % difference in PM2.5 exposure among racial-ethnic groups per MWH generation over time for each scenario, or providing other information regarding how changes in disparities over time can be attributed to changes in

emissions/MWh vs changes in total generation, population counts, etc, could help tell the story more clearly. Also, by only showing 2020, 2035, and 2050 for many of the figures, it is difficult to tell exactly when equality is achieved in different scenarios, which seems important.

Some other specific comments:

Line 33: air -> air

Line 91: population -> pollution

Figure 1: Coal and Coal-CCS seem to be the same color

Throughout: The distinction between emissions and concentrations of PM2.5 and its precursors could be made clearer and more consistent. Usually powerplants release "emissions" but people are exposed to "concentrations".

Figure 4: Do these values represent just the average of the census tracts, or is it a population-weighted average? (Different census tracts have similar populations, but they're not all the same)

Figure 5: The figure caption says PM2.5 but then talks about NOx and SO2. Is it meant to be secondary PM2.5 concentrations resulting from emissions of NOx and SO2? Also, as above, it is important to make clear distinctions between emissions and concentrations.

Table 2: Metric units rather than pounds per mmbtu?

Line 428: It's important where they end up as ambient concentrations in the air rather than where they settle

Figures s10-s14: As above, are these concentrations of NOx and SO2, or concentrations of secondary PM2.5 caused by emissions of NOx and SO2?

Reviewer #3 (Remarks to the Author):

Please see my detailed comments for the authors in the attached file. -Dallas Burtraw

Reviewer #3 (Remarks to the Author):

April 25, 2022

Comments on: Inequality in energy transitions: Air pollution disparities amidst national decarbonization strategies

This manuscript evaluates alternative pathways for decarbonizing the electricity sector that proceed at different speeds, using different instruments, and arriving at different endpoints. The manuscript finds that the pollution burden falling on disadvantaged communities remains greater than for other communities until full decarbonization is achieved.

The manuscript is hampered by the challenge of comparing scenarios where more than one thing is changing at the same time. Hence, direct comparison between the scenarios is usually not convincing. The authors have addressed this by seeking “national energy equality.”

It is profoundly important and intellectually challenging to robustly address the policy goals and strategies that might be associated with different outcome measures. The failure to achieve national energy equality at an interim milestone before the endpoint of zero fossil fuel combustion is achieved does not indicate how the greatest benefits can be achieved at the greatest pace for the greatest number of people, and especially for disadvantaged communities. The contemporary political narrative of environmental justice is served by retaining a focus on direct comparison of outcomes across the population, as the authors do, but the manuscript would be improved if it also evaluated the pathways in terms of the most rapid progress for specific communities and for the nation, and if it evaluated the pathways which close the gap among community-level outcomes the most quickly (even before complete equality is achieved). Disadvantaged communities start out at severe excess burden compared to their counterparts. In this context, *any benefit* to an “advantaged” community paradoxically widens rather than narrows the gap and undermines progress according to the author’s applied metric.

In general, the manuscript would be improved if it would provide a ratio across communities and populations of interest and describe the changes in the ratio, that is, how each policy proportionately affects each community.

Another metric that would be quite valuable would be to report the ratio of PM to CO₂ according to different population categories over time so that we could observe which decarbonization policy leads to the greatest ton-for-ton improvement in health outcomes. Another metric would be to describe PM/MWh. The important thing that is missing overall is to describe the annual rate of change in reported measures.

The authors do not have a table summarizing and describing the scenarios until it appears in the methods section. However, it is impossible to describe present the results without labeling the scenarios. The labels are not sufficient, and burden is placed on the reader to keep things straight based on the summary in the narrative. Hence, elements or all of the scenario table should be brought up in the manuscript to appear within the results section. This breaks “the rules” for manuscript organization but rules are meant to be broken when it serves purpose. The editor will need to weigh in on this.

Not until the very end of the paper is the reader reminded that costs of emissions mitigation may fall unevenly across the population. Indeed, as a share of income, the disadvantaged populations may bear a disproportionate cost which could have adverse health outcomes of comparable magnitude as

examined here. I do not want to undermine the analysis of the benefits of air quality improvements, but this is more than a minor caveat and I believe it should be mentioned in the introduction.

14-17: The reader might infer that black or poor communities have higher pollutant concentrations during the transition than they do in the baseline.

33: “aur”

57: “yet...” I am not sure the element of surprise that is implied by this word choice. It would be sufficient to directly state “and...”

85: Another reduced complexity paper that should be cited at the outset is a report that evaluates economy wide impacts and looks at the distribution of air quality benefits across demographic groups in a very similar way. See <https://www.rff.org/publications/reports/the-distribution-of-air-quality-health-benefits-from-meeting-us-2030-climate-goals/>

93: “investigated researched”

208: “depositing” Air pollution is not deposited in communities, unless one is referring to acidification of the environment or other ecological outcomes. Air pollution concentrations change and are transitory over geography. It would be better to write “...through the atmosphere affecting pollutant concentrations and causing health impacts.”

271-278: There may be deeper concerns than what is listed in this paragraph. Cumulative exposures combined with other community-level stressors point to different concentration-response relationships. See Spiller et al. Mortality Risk from PM2.5: A Comparison of Modeling Approaches to Identify Disparities Across Ethnic Groups in Policy Outcomes.

281: The legend should indicate that this is change in average PM concentration associated with changes in power generation.

310: Black communities see the highest emissions. They also see the greatest reductions.

316: “...historical trends will exacerbate health impacts...” This may be true (see Spiller reference above) but cannot be asserted without a reference or supporting evidence.

374: Does ReEDS achieve dynamic consistency? Is there a chance that an investment is made in, say, 2025 that is regrettable come 2030? This should be checked.

401: What is the relevance of CO2e as a metric given that only combustion related CO2 is used to evaluate the climate policy?

407: I believe that Table 2 should be reported in pound/MWh. This will have much greater relevance for a broad community and make the paper more widely cited.

Air pollution disparities and equality assessments of US national decarbonization strategies

Response to reviewer comments*

***Reviewer comments are indicated in black, our responses in blue, and our text additions in red. References are at the end of the document.**

Reviewer 1

This paper examines how decarbonization pathways driven by a least-cost paradigm will impact air pollution inequality across different vulnerable groups in the United States. A key finding is that Black or poor communities have higher co-pollutant concentrations during the energy transition, until a national mandate requires full deployment of renewable or low-carbon technologies. The questions posed by the authors are important and policy-relevant. However, several aspects (including the method to obtain the distribution of air pollution at regional scale and across vulnerable groups) should be explained more carefully before the paper can be considered for publication. Some suggestions are offered below:

Thank you for your feedback to help improve our paper. We appreciate the time and input of your review. Please find our responses to your comments below.

- In the Introduction:
 - You add to the literature by evaluating environmental sustainability and equality across eight national decarbonization pathways between 2010 and 2050. Is your paper the first one taking a forward-looking approach for this type of analysis? Or are there other papers that evaluate future distributional effects in connection to an energy systems model, and how does your analysis differ from theirs?

We acknowledge that there are other papers which investigate trade-offs at the national scale, our paper is novel in the sense that it looks at a sub-national scale, and ties national energy transitions to census tract impacts. There are other papers that investigate the distributional effects of energy systems models at low-spatial resolutions, meaning they may miss sub-national disparities. Others that do have more granular spatial resolutions look at limited decarbonization scenarios or do not investigate the differences in impact across different demographic groups.

- Sasse & Trutnevyte (2020) investigate the country-level trade-offs of different objective priorities, like cost, equality, or renewable energy generation in Europe in 2035 (Sasse & Trutnevyte, 2020). They use a modeling to generate alternatives (MGA) approach to run 100 different generated alternatives. Our analysis differs from this because we look at air pollution disparities at the census tract level across the entire US. We also use a least-cost optimization without an MGA approach.

- Sergi et al. (2020) incorporate air pollution damages and social costs into the objective function of a capacity expansion model to investigate how implementing a multi-objective view of energy planning may impact investments (Sergi et al., 2020). Our paper differs from this because we investigate the differences in air pollution exposure across different demographics and do not include the cost of air pollution damages in the objective function of our capacity expansion model.
- Dimanchev et al. (2019) investigate the co-benefits of different decarbonization policies on the rust belt in the US (Dimanchev et al., 2019). Our work expands on this because it looks at the impact of different decarbonization policies on census tracts across the entire US.
- Luo et al. (2021) investigate the air pollution benefits across different demographic groups in Texas by internalizing health impacts in energy planning across two decarbonization scenarios (Luo et al., 2021). Our work expands on this by doing a national-scale investigation of air pollution impacts on different demographic groups.
- Mayfield (2022) investigate the air pollution impacts on mortalities based on future coal plant retirements under two scenarios (no policy change and net zero) and across different demographic groups using a multi-objective energy and equity model (Mayfield, 2022). This work only focuses on coal retirements, rather than national scale decarbonization efforts and renewable energy investments as well as focusing on only two retirement scenarios.
- Burtraw et al. (2022) use a capacity expansion model tied to a reduced complexity model to investigate the air pollution benefits of reaching climate goals by 2030 (Burtraw et al., 2022). Our work uses a similar framework by tying a capacity expansion model to a reduced complexity air pollution model. They also investigate at a high spatial resolution (county level) and across different demographic groups. We build on this work by looking at the impact of many decarbonization policies on equality of air pollution across the US.

We have updated the text in the main document to capture these analyses. See for example the following text in the main:

[Lines 92-122] “While least-cost optimization models often exclude equality considerations, some papers have integrated equality and distributional analysis into the electricity system decision making paradigm. One paper investigated the social and environmental implications of expanding power systems in developing countries with little to no existing infrastructure (Nock et al., 2020) at a subnational level. Sasse & Trutnevte (2020) investigated the sustainability and equity impacts of reaching electricity sector targets across European countries. While this paper highlights four different optimization objective scenarios (base case, cost, equality, and renewable generation), their focus is on intercountry equality considerations, which miss local level equality impacts. Sergi et al. (2020) does a forward-looking analysis to investigate the impact of including co-benefits of decarbonization by including damages from air pollution in the objective function, but they do not explore distributional energy justice or equality of air pollution across different decarbonization scenarios. Dimanchev et al. (2019) investigate the co-benefits of different decarbonization policies on the rust belt and the regional distribution of these co-benefits but also does not investigate the impacts on different demographics. Luo et al. (2021) investigate the air pollution benefits across different demographic groups in Texas by internalizing health impacts in energy planning. Mayfield (2022) investigates the air pollution impacts on mortalities based on future coal plant retirements under two scenarios (no policy change and net zero) and across different demographic groups using a multi-objective energy

and equity model. Burtraw et al. (2022) use a capacity expansion model tied to a reduced complexity model to investigate the air pollution benefits of reaching climate goals by 2030 across counties and different demographic groups. We build on this work by investigating how least-cost optimization (dominant decision paradigm) for energy planning in the US impacts local equality objectives across eight decarbonization scenarios, some of which include 80-100% renewable penetration, national carbon caps, or 100% low carbon technology requirements. From this analysis, we gain policy insights on how different decarbonization scenarios will impact equality goals throughout the energy transition while also quantifying how different groups will be burdened with air pollution.”

- You only cite a couple of papers considering historic distributional effects, but in fact this literature is growing (see, among others, the recent work by Erin Mayfield and Ines Azevedo). Please add a more detailed review of the existing literature focusing on historic distributional effects.

Thank you for this feedback. We have built on our discussion of literature that considers the historic distributional effects of air pollution.

[Lines 67-80] “Some studies have investigated the air quality co-benefits of decarbonization policies (i.e., additional reductions in other emissions like PM_{2.5}). In general, mitigating greenhouse gas emissions results in positive co-benefits in PM_{2.5} emissions (Saari et al., 2015; Thompson et al., 2014; Zhang et al., 2017) at the system level, but there is still uncertainty regarding the spatial distribution of these emission reduction benefits especially within vulnerable communities. Multiple studies have investigated the air pollution exposure disparities across different racial and income groups using retroactive analyses, finding that low-income, Black, Asian, and Hispanic or Latinx communities were exposed to higher levels of PM_{2.5} in the US in 2000, 2014, and 2016 (Jbaily et al., 2022; S Thind et al., 2019), which stem from historical policy inequities (Lane et al., 2022). While it is valuable to understand the level of historical injustices, countries also need a framework to that evaluates future disparities in air pollution distributions across the nation under different decarbonization plans in order to mitigate and reduce future inequalities. Here we create a forward-looking analysis framework for assessing how decarbonization benefits will be shared across different demographic groups by tying a capacity expansion model with a local equality analysis.”

- The paper would benefit from a better description of the scenarios. For example:

We have moved the description of the scenarios to the beginning of the results to make the presentation of our results clearer and have added a more detailed description of the decarbonization scenarios into the paper, For example the table of the decarbonization scenarios has been moved, and we added some more text to the results. See example excerpt below:

[Lines 143-158] “The base case (Scenario A) assumes all current carbon and energy policies remain in place, like state renewable portfolios, the Cross-State Air Pollution Rule, investment and production tax credits, and regional cap-and-trade policies. All other scenarios include these policies as well. The carbon cap scenarios were defined by estimations in CO₂ reductions from the Energy Innovation Technology & Policy LLC modeling to US Nationally Determined

Contributions from the 2015 Paris Agreement (Scenario B) and emissions reductions required to keep global warming below 1.5°C (Scenario C) (Energy Innovation Policy & Technology LLC, n.d.). The Energy Innovation Technology & Policy LLC’s model that simulates decarbonization scenarios deploys a mix of different policies to reduce emissions from all sectors. We focus on the electricity sector and use their estimations in reductions in emissions to reach either US nationally determined contributions (NDC) or the 1.5C pathway. Scenario B, which implements a carbon cap based on US NDC allows for an increase in carbon emissions from 2040 to 2050 due to the policy assumptions made by the Energy Innovation Technology and Policy LLC modeling tool. The national technology mandate scenarios (Scenarios D-H) were defined by assuming renewable energy and low carbon deployment began at 20% in 2020 and linearly increased to the mandate year (either 2035 or 2050).

- How do you obtain the annual emissions under the 2 carbon cap scenarios (B and C)?

The carbon cap scenarios were obtained from the Energy Policy Solutions model (version 3.1.0) from the Energy Innovation Technology & Policy LLC for scenarios of emissions reductions given either US NDC or 1.5C Pathway constraints. Their model deploys a mix of different policies to reduce emissions from all sectors. We focus on the electricity sector and use their estimations in reductions in emissions to reach either US nationally determined contributions (NDC) or the 1.5C pathway.

We have added a statement in the text to clarify this:

[Lines 146-153] “The carbon cap scenarios were defined by estimations in CO₂ reductions from the Energy Innovation Technology & Policy LLC modeling to US Nationally Determined Contributions from the 2015 Paris Agreement and emissions reductions required to keep global warming below 1.5°C (Energy Innovation Policy & Technology LLC, n.d.) . The Energy Innovation Technology & Policy LLC’s model that simulates decarbonization scenarios deploys a mix of different policies to reduce emissions from all sectors. We focus on the electricity sector and use their estimations in reductions in emissions to reach either US nationally determined contributions (NDC) or the 1.5°C pathway.”

- What specific technologies/sources are included in the RPS targets in scenarios D-F?

The technologies included in these scenarios include solar photovoltaics (PV), onshore wind, offshore wind, concentrated solar power (CSP), biopower, hydropower, geothermal, landfill gas, pumped hydropower storage, and battery storage. We have added a sentence in the paper to clarify this:

[Lines 158-162] “The technologies included in the national renewable energy mandate scenarios (Scenario D: 80% RE by 2050, Scenario E: 100% RE by 2035, and Scenario F: 100% RE by 2050) are solar photovoltaic (PV), concentrated solar power (CSP), onshore and offshore wind, biopower, hydropower, geothermal, landfill gas, pumped hydropower storage and battery storage.”

- The Low Carbon scenarios reach “100% renewable energy, natural gas CCS, or nuclear”. What specific renewable energy technologies/sources are included? Are RE/gas+CCS/nuclear mutually exclusive, or can they be used jointly to reach the target?

For renewable energy, solar photovoltaics (PV), onshore wind, offshore wind, concentrated solar power (CSP), biopower, hydropower, geothermal, and battery storage are included. Renewable energies, natural gas CCS, and nuclear can be used jointly to reach the specified target. We have clarified this in the paper:

[Lines 162-165] “The national low carbon mandates (Scenario G: Low Carbon by 2035 and Scenario H: Low Carbon by 2050) allow generation from the same renewable energy technologies specified in the national renewable energy mandates, as well as generation from natural gas carbon capture and storage (CCS) and nuclear power plants.”

- Scenario B sees an increase in all emissions and coal generation from 2040 to 2050. Why?

We acknowledge that it is odd that the carbon cap used in Scenario B allowed an increase in CO₂ emissions from 2040 to 2050. The carbon cap scenarios were based on the Energy Policy Solutions modeling (version 3.1.0) from the Energy Innovation: Technology and Policy LLC (Energy Innovation Policy & Technology LLC, n.d.). This may be due to the policies that the Energy Innovation Policy & Technology LLC used to reduce emissions in the electricity sector. We have added this reference into the main text.

[Lines 153-155] “Scenario B, which implements a carbon cap based on US NDC, allows for an increase in carbon emissions from 2040 to 2050 due to the policy assumptions made by the Energy Innovation Technology and Policy LLC modeling tool.”

- Figure 4: Looking at the solid red line (Black), can you explain why average PM_{2.5} concentrations rise until ~70%, and then decrease? Can you explain the trend in the dotted blue line (White)? Results for other minority groups (in particular, Latinx/Hispanic) seem to suggest that average concentrations of PM_{2.5} decrease as the share of Latinx/Hispanics increases – isn’t this a counterintuitive results?

Thank you for presenting counter intuitive results. We have gone back and redone the analysis. We have replaced Figure 4 to show population weighted values across the years 2020 to 2050. This shows disparities between racial groups over the energy transition in a more direct way. The population weighted figures produce similar trends across race and ethnicities with Black people exposed to the highest concentrations of air pollution, followed by non-Latinx white people, and then with Latinx or Hispanic, Asian, and Indigenous people exposed to the least air pollution from electric generating units (EGUs). See Figure 4 below.

Research on the current burden of air pollution from EGUs in the US shows similar trends: Black people are exposed at the highest concentrations of PM_{2.5} and non-Latinx white people have the

second highest exposure rates (S Thind et al., 2019). The disparity seen in Latinx and Hispanic, Asian, and Indigenous people compared to Black and non-Latinx white people may be due to coal plants built in rural areas with higher white populations. Also, we find that the percent change in concentrations from 2020 to 2050 across demographics are similar, indicating that the bulk of disparities in the electricity sector likely stem from historical inequities in power plant siting and legacy generation rather than future investments. This has important implications for the retirement strategy for legacy generation.

Another reason the results are showing lower exposure among Latinx/Hispanic, Asian, and Indigenous people is because we are investigating emissions and air pollution concentrations only from the electricity sector. Our analysis does not capture air pollution injustices caused by transportation, industrial, or residential cooking activities. Research has shown that air pollution from these activities is where Hispanic and Asian communities are disproportionately affected (Tessum et al., 2021). Li et al. (2022) shows that Indigenous communities are also disproportionately burdened by PM_{2.5} concentrations when accounting for all emission sources (Li et al., 2022). We have added text into the paper to discuss this.

[Lines 332-341] “Figure 4 also shows that Latinx or Hispanic, Asian, and Indigenous people in the US are exposed to less PM_{2.5}, on a population weighted basis, from power plants in the electricity sector. This is consistent with historical trends where Black and non-Latinx white communities have been disproportionately impacted by coal fueled power plants (S Thind et al., 2019). Our scope is limited to the investigating emissions and air pollution concentrations from the electricity sector. However, we note that air pollution injustices are also caused by transportation, industrial, or residential cooking activities, which have disproportionate impacts on Asian, Hispanic, (Tessum et al., 2021), and Indigenous communities (Li et al., 2022).”

[Lines 337-344] **Figure 4:** Population weighted average annual PM_{2.5} concentrations (in µg/m³) across different race and ethnicity groups for each decarbonization scenario from 2020 to 2050. We see that Black communities in the US are exposed to higher concentrations of PM_{2.5} in

2020, which is consistent with historical impacts (S Thind et al., 2019). Over the energy transition, Black communities are exposed to higher concentrations of PM_{2.5} until a technology mandate is >80% renewable energy (Scenario D in 2050, Scenario E in 2035, or Scenario F in 2050), 100% low carbon energy (Scenario G in 2035 and Scenario H in 2050), or carbon cap that requires emissions of CO₂ less than 400 Mt (Scenario C in 2030).”

- Figure 5:
 - You write: “The points on the plot represent the minimum median income in each group (e.g., the point at \$0k represents the <\$25k income group).” Please clarify this statement. For example, what do the points at \$50k, \$100k and \$150k represent?

The original points were representative of median income groups by \$25,000, with the point at \$0k representing census tracts with median incomes below \$25k, the point at \$25k representing census tracts with median incomes between \$25k-\$50k, etc. However, we recognize that displaying information this way may be unclear. Therefore, we have updated the figure to look at population weighted air pollution concentrations across different median income groups. The income groups are defined by different colored lines in the graph, instead of along the x-axis. Figure S-12 displays this change below. Because the income groups have similar concentrations across the energy transition, we have also shown the highest (>\$150k), lowest (<\$25k), and \$100k-\$125k median income groups in their own figure to see the differences more clearly (Figure 6 below). We find that across income groups the difference of population weighted PM_{2.5} and NO_x concentrations is a maximum of 0.22 µg/m³. Therefore, disparities among income groups for PM_{2.5} and NO_x are minimal.

“Figure S-12: Population weighted annual average PM_{2.5} across median income groups in each scenario and years 2020 – 2050.”

[Lines 386-389] **Figure 6:** Distribution of population weighted annual average PM_{2.5} across each scenario 2020 to 2050 for the highest (>\$150k), mid (\$100k-\$125k), and lowest (<\$25k) income groups. The other income groups fall in between the highest and lowest bounds. Results

from all scenarios can be found in SI Figure S-12, and NO_x and SO₂ across income groups in Figures S-13 and S-14 respectively.”

- Why are some pollutants (e.g., NO_x, PM_{2.5}) increasing as median income increases?

When we calculate the population weighted air pollution concentrations for median income groups (Figures S-12, S-13, and S-14 in the SI), we find that NO_x concentrations are highest in the lowest (<\$25k) and highest (>\$150k) income groups (Figure S-13). This could be because of location of different plants that have comparable NO_x emissions rates to coal plants: natural gas combustion turbine has a NO_x emissions rate of 0.637 g/kWh versus coal plants with 0.672 g/kWh. Another reason could be higher median incomes are also located near cities. This is a limitation of using gross median income since it does not account for cost of living. However, we can capture cost of living differences across census tracts through poverty rates. When looking at poverty rates, we see that census tracts with >70% of poverty are exposed to the highest concentrations of co-pollutants.

While there is not much difference between the income groups in PM_{2.5} and NO_x concentrations, we find that there is a large disparity between the highest and lowest income group in population weighted SO₂ concentrations (SI Figure S-14). This disparity is likely due to coal plant placement in low-income areas (Tarekegne et al., 2021). Coal power plants are the main contributor to SO₂ emissions, as seen in SI Figure S-5. We have added this discussion about the distribution of air pollution across median income into the SI and moved the discussion of distribution across poverty groups into the main text.

Overall, we find that median income is less of an indicator of population weighted air pollution exposure than race, ethnicity, or poverty within a region. We have investigated the exposure of air pollution across race and ethnicity groups within each median income group. From this, we find that within each median income group, Black communities are exposed to higher concentrations of population weighted air pollution than any other race or ethnicity group. We have included these findings in figures below and in the SI Figure S-15. We have updated our main text as follows:

[Lines 376-384] “Overall, we find that median income is less of an indicator of population weighted air pollution exposure than race, ethnicity, or poverty within a region. Within each median income group, Black communities are exposed to higher concentrations of population weighted air pollution than any other race or ethnicity group (see SI Figure S-15).

PM_{2.5} and NO_x concentrations across income groups do not see significant differences, but we find that there is a large disparity between the highest and lowest income group in population weighted SO₂ concentrations (SI Figure S-15). This disparity is likely due to coal plant placement in low-income areas (Tarekegne et al., 2021). Coal power plants are the main contributor to SO₂ emissions, as seen in SI Figure S-5.”

“Figure S-13: Population weighted NO_x concentrations across income groups and scenarios 2020 – 2050.”

“Figure S-14: Population weighted SO₂ concentration (µg/m³) for each scenario across median income groups. We see that the lowest income group (<\$25k) has the highest concentration of SO₂ and the highest income group (>\$150k) has the lowest concentration until technology mandate of low carbon is >80% (Scenarios D-H) or a carbon cap requires less than 400 Mt (Scenario C).”

“Figure S-15: Distribution of population weighted PM_{2.5} concentration across race and ethnicity in the two highest (\$125k-\$150k and >\$150k) and two lowest (<\$25k and \$25k-\$50k) median income brackets for Scenario A (Base Case) and Scenario D (80% RE by 2050 Mandate). This

shows that across income groups, Black people are exposed to the highest concentrations of PM_{2.5}.”

- Please describe the InMAP model in more detail. You are currently referring to the model documentation for details, but the reader should be able to understand important features of the model used in your analysis without having to consult other papers.

Thank you for this insight. The InMAP model intakes a shapefile that specifies emissions at any regional level (for us, we use the ReEDS region level). Emissions are then area weighted across the ReEDS region based to fit the InMAP defined regions. InMAP uses these area weighted emissions to estimate the average annual concentration across a spatial resolution of square regions sized 1 to 48 kilometers wide.

We have added more explanation of the InMAP model into the paper in SI Section C as follows:

[SI section referenced in Line 538 in main text] “InMAP intakes emissions at the ReEDS level and area weights them across the ReEDS region to fit InMAP defined regions (1x1 km to 48x48 km squares). InMAP uses these emissions input to estimate the average annual concentration. While running, InMAP uses a reaction-advection-diffusion equation that estimates where air pollution ends up as ambient concentrations. The model uses a steady state formulation for each time step and continues to run until the air pollution concentrations reach steady state (the change in concentration is zero). Within each time step, each region accounts for the flux of new emissions and how pollution concentrations are affected by physical and chemical processes. Once the model reaches steady state, it outputs a shapefile with the annual average ambient concentrations for each region. (Tessum et al., 2017)”

- I would like to see a better description of the method to obtain the distribution of air pollution at regional scale and across vulnerable groups. This is a key part of your analysis, and should be discussed more carefully to help the reader understand the drivers of your results.

The distribution of air pollution across vulnerable groups was obtained using census tract data tied to air pollution estimates at the census tract level.

We have added a more detailed explanation of how we obtained the distribution across census tracts and demographics, as well as Figure 6 (below), which shows how the results were obtained from the air pollution model.

[Lines 538-540] “InMAP was tied to census tracts by taking the average concentration across modeling regions for each census tract, as detailed in Figure 6 below.”

- Please describe how the ReEDS and InMAP models are linked in more detail.

Thank you for highlighting where we could add more detail. To tie ReEDS to InMAP, shapefiles of emissions (NO_x, SO₂, and PM_{2.5}) at the ReEDS region level are the primary input into InMAP.

We have updated the text to include more detail about the inputs of the models. Once these emissions from ReEDS are input into InMAP, the model area weights the emissions across the ReEDS regions to the InMAP regions. We display this downscaling in Figure 6 in the paper (as seen below). The area weighting equation can be found in SI Equation S-10. Below we detail what we have in SI Section C about the area weighting now:

“InMAP uses area-weighting to distribute emissions at the ReEDS level (134 regions) to the InMAP level (squares with 1 to 48 km sides). Equations S-1a and S-1b display the two-step process for area-weighting (Prener, n.d.). In Equation S-1a, the areal weight for each InMAP region is calculated, where $W_{I,i}$ is the areal weight for the InMAP region i (in km^2), $A_{I,i}$ is the area of the InMAP region (in km^2), and $A_{R,j}$ is the area of the ReEDS region j . Equation S-1b calculates the areal weighted air pollution value in each InMAP region, where $E_{I,i}$ is the estimated air pollution magnitude in each InMAP region and $Q_{R,j}$ is the PM, NO_x , or SO_2 air pollution (in kilograms) in each ReEDS region j .

$$W_{I,i} = \frac{A_{I,i}}{A_{R,j}} \quad (\text{Eq. S-1a})$$

$$E_{I,i} = W_{I,i} * Q_{R,j} \quad (\text{Eq. S-1b})$$

Once the emissions are area weighted, InMAP begins its modelling of air pollution transport and chemical reactions. In the paper, we have updated Figure 7 to include diagram to clarify how the models are linked:

[Lines 556-560] “Figure 7: Equality analysis methods using a capacity expansion model (ReEDS) and a reduced complexity air pollution transport model (InMAP) to investigate where emissions are settling after being emitted from power plants. Regions are downscaled from ReEDS regions to census tract level and annual average air pollution concentrations are estimated from emissions at the ReEDS region level.”

- Some typos – please proofread carefully before submission.

Thank you. We have carefully read the paper and corrected the typos.

Reviewer 2

This article provides important information regarding equity implications of energy transition scenarios. This is an important topic, the methods used are scientifically rigorous, and the results support the claims that are made. The methods are well described.

We thank you for your feedback and comments to improve our paper. It is very appreciated. Please find our responses to your comments below.

I feel that the paper could be made even better by going a little further in the analysis of the data. For example if there were plots showing the absolute or % difference in PM_{2.5} exposure among racial-ethnic groups per MWh generation over time for each scenario, or providing other information regarding how changes in disparities over time can be attributed to changes in emissions/MWh vs changes in total generation, population counts, etc, could help tell the story more clearly.

In the figures below we show the absolute and percent changes in PM_{2.5} exposure 2020 to 2050 across race and ethnicity. We see that Black people have the largest absolute change in PM_{2.5} exposure, followed by non-Latinx white people. While the magnitude of change across race and ethnicity varies, the percent change of emissions across race and ethnicity groups are almost identical within a scenario, which indicates that the starting point of air pollution concentrations is a significant indicator of future inequalities. This means that historical injustices will continue throughout the energy transition until widespread deployment of low carbon technologies is reached. We have also done these investigations based on PM_{2.5} concentration per MWh of generation at the national level, also seen below and in SI Figure S-7.

While we believe it would be interesting to look at PM/MWh distribution across racial groups, we find this may convolute our analysis. Emissions from power plants do not always settle close to those power plants (Tessum et al., 2017), so it may be the case that different groups are affected by more generation that is not in their region. Thus, we believe the PM/MWh ratio may be misleading at the census tract level. We believe that this is valuable at the national level, and we display this ratio in Figure S-7 in SI.

We include the following text in the main article to refer to the finding of the SI Section G:

[Lines 307-311] “Based on these results, Black communities are at risk for higher PM_{2.5} concentrations and its associated health impacts in energy transitions but also see the largest absolute reductions in air pollution exposure (see SI Figure S-16). The percent change of PM_{2.5} concentration across race and ethnicity changes at the same rate within each scenario (see SI Figure S-17), indicating that the starting point of air pollution impacts future exposure throughout the energy transition.”

“Figure S-16: Absolute change of PM_{2.5} concentrations across race and ethnicity groups.”

“Figure S-17: Percent change of PM_{2.5} emissions across race and ethnicity groups.”

Also, by only showing 2020, 2035, and 2050 for many of the figures, it is difficult to tell exactly when equality is achieved in different scenarios, which seems important.

We have added in the midterm years (2026, 2030, 2040, and 2045) of the model runs to the air pollution concentration across different demographics (Figures 4 and 5, for example) to see more clearly when equality is achieved in different scenarios. We believe the line graphs help us accomplish this point. We have opted to keep the maps for the years 2035 and 2050 due to the mandate year policies because our objective was to show how the decarbonization policies led to different outcomes.

Some other specific comments:

Line 33: aur -> air

Line 91: population -> pollution

Figure 1: Coal and Coal-CCS seem to be the same color

Thank you for pointing these out. We have made the corrections in the text.

In Figure 1, we removed Coal-CCS from the legend because it is never built due to its high price relative to other technologies. We have attached the updated figure below (Figure 1 in paper).

[Lines 206-210] “Figure 1: Annual generation mix (PWh) 2010 – 2050 by technology for each decarbonization scenario resulting from the ReEDS model. We highlight that the renewable and low carbon technology mandates accommodate additional energy needs primarily through expanded wind and solar generation investments. We see that the base case, US NDC, and 80% renewable energy decarbonization pathways retain coal generation through 2050.”

Throughout: The distinction between emissions and concentrations of PM2.5 and its precursors could be made clearer and more consistent. Usually powerplants release “emissions” but people are exposed to “concentrations”.

Thank you for this feedback. We have edited instances in the paper to make this distinction clearer. We have also added a clarifying statement in the text regarding the distinction between PM2.5 and its precursors. Text excerpt:

[Lines 250-252] “PM_{2.5} concentrations throughout results include both primary PM_{2.5} (directly from the power plant) and secondary PM_{2.5} (formed from NO_x and SO₂). NO_x and SO₂ results are actual concentrations of NO_x and SO₂, not secondary PM_{2.5} formed by these pollutants.”

Figure 4: Do these values represent just the average of the census tracts, or is it a population-weighted average? (Different census tracts have similar populations, but they're not all the same)

We appreciate your feedback and have done a population weighted average concentration across groups. We have included our updated Figure 4 below:

[Lines 343-350] **Figure 4:** Population weighted average annual PM_{2.5} concentrations (in µg/m³) across different race and ethnicity groups for each decarbonization scenario from 2020 to 2050. We see that Black communities in the US are exposed to higher concentrations of PM_{2.5} in

2020, which is consistent with historical impacts (S Thind et al., 2019). Over the energy transition, Black communities are exposed to higher concentrations of PM_{2.5} until a technology mandate is >80% renewable energy (Scenario D in 2050, Scenario E in 2035, or Scenario F in 2050), 100% low carbon energy (Scenario G in 2035 and Scenario H in 2050), or carbon cap that requires emissions of CO₂ less than 400 Mt (Scenario C in 2030).”

Figure 5: The figure caption says PM_{2.5} but then talks about NO_x and SO₂. Is it meant to be secondary PM_{2.5} concentrations resulting from emissions of NO_x and SO₂? Also, as above, it is important to make clear distinctions between emissions and concentrations.

The PM_{2.5} concentrations shown in the original Figure 5 were total PM_{2.5}, which is both primary and secondary PM_{2.5}. We have clarified this in the paper. We supply the results for PM_{2.5}, NO_x, and SO₂ concentrations across median income groups in SI Figures S-12, S-13, and S-14 respectively.

Table 2: Metric units rather than pounds per mmbtu?

We have put Table 2 into metric units as seen below. The original table was put in SI Table S-3 along with the heat rates used to convert to g/kWh.

[Lines 503-507] Table 2: Operating emission rates used in environmental sustainability analysis [in g/kWh]. See Table S-2 in SI for sources.*

	CO ₂ [g/kWh]	NO _x [g/kWh]	SO ₂ [g/kWh]	PM _{2.5} ** [g/kWh]
Biopower	0	0	0.490	0.620
Solar photovoltaic (PV)	0	0	0	-
Concentrated solar power (CSP)	0	0	0	-
Onshore wind	0	0	0	-
Offshore wind	0	0	0	-
Nuclear	0	0	0	-
Natural gas combustion turbine (CT)	496	0.637	0.064	0.028
Natural gas combined cycle (CC)	337	0.058	0.015	0.019
Natural gas CCS	39.8	0.069	0.017	0.022
Hydropower	0	0	0	0
Geothermal	0	0	0	0
Oil-Gas-Steam	662	0.832	1.44	0.081
Coal	923	0.672	2.06	0.069 ¹
Coal IGCC	756	0.305	0.199	0.056 ¹
Coal CCS	97.0	0.392	0.256	0.072 ¹

Cofire	821	0.598	1.89	0.131
Battery storage	0	0	0	0
Pumped hydropower	0	0	0	0

Line 428: It's important where they end up as ambient concentrations in the air rather than where they settle

We have changed the wording in the paper to address this.

[Lines 526-529] “To understand how energy transitions may exacerbate or reduce local air pollution disparities, we use a reduced complexity model, InMAP, to quantify where co-pollutants (NO_x, SO₂, and PM_{2.5}) end up as ambient concentrations after being emitted from power plants.”

Figures s10-s14: As above, are these concentrations of NO_x and SO₂, or concentrations of secondary PM_{2.5} caused by emissions of NO_x and SO₂?

We have focused on including the co-pollutants because these are all reported to have health impacts (Kampa & Castanas, 2008). In the original Figures S10-S14 (that have now been replaced with population weighted results), these are concentrations of NO_x and SO₂. The concentrations of secondary PM_{2.5} are captured in the PM_{2.5} results because we use total PM_{2.5} (which is primary PM_{2.5} plus secondary PM_{2.5}). We have clarified this in the text as follows:

[Lines 247-249] “PM_{2.5} concentrations throughout results include both primary PM_{2.5} (directly from the power plant) and secondary PM_{2.5} (formed from NO_x and SO₂). NO_x and SO₂ results are concentrations of NO_x and SO₂, not secondary PM_{2.5} formed by these pollutants.”

Reviewer 3

This manuscript evaluates alternative pathways for decarbonizing the electricity sector that proceed at different speeds, using different instruments, and arriving at different endpoints. The manuscript finds that the pollution burden falling on disadvantaged communities remains greater than for other communities until full decarbonization is achieved.

Thank you for taking the time to review and give feedback to our paper. It is tremendously appreciated. We have addressed each of your comments below.

The manuscript is hampered by the challenge of comparing scenarios where more than one thing is changing at the same time. Hence, direct comparison between the scenarios is usually not convincing. The authors have addressed this by seeking “national energy equality.”

It is profoundly important and intellectually challenging to robustly address the policy goals and strategies that might be associated with different outcome measures. The failure to achieve

national energy equality at an interim milestone before the endpoint of zero fossil fuel combustion is achieved does not indicate how the greatest benefits can be achieved at the greatest pace for the greatest number of people, and especially for disadvantaged communities. The contemporary political narrative of environmental justice is served by retaining a focus on direct comparison of outcomes across the population, as the authors do, but the manuscript would be improved if it also evaluated the pathways in terms of the most rapid progress for specific communities and for the nation, and if it evaluated the pathways which close the gap among community-level outcomes the most quickly (even before complete equality is achieved). Disadvantaged communities start out at severe excess burden compared to their counterparts. In this context, any benefit to an “advantaged” community paradoxically widens rather than narrows the gap and undermines progress according to the author’s applied metric.

Thank you for this perspective on how to best highlight the equality value of different decarbonization pathways. In the main text we have reframed some of our figures to focus on the population weighted disparities between different racial groups. We have also included a figure on the absolute changes in PM_{2.5} concentrations across racial groups (Figure S-17 in the SI) and we have added a sentence on which scenario shirks the disparity gap between racial groups the fastest, as seen below:

[Lines 311-315] “We find the 1.5C pathway carbon cap and 100% renewable energy deployment by 2035 decarbonization pathways (Scenarios C and E respectively) shirks the inequality between racial groups the fastest (by 2030). This is important for recognition justice because disadvantaged communities start out at severe excess burden compared to their counterparts (S Thind et al., 2019) Scenario E has the highest absolute reductions in PM_{2.5} concentrations across racial groups (SI Figure S-16).”

In general, the manuscript would be improved if it would provide a ratio across communities and populations of interest and describe the changes in the ratio, that is, how each policy proportionately affects each community.

Thank you for highlighting how we can make our results clearer. We address this in a few ways. 1) We now present our results in the population weighted concentration of co-pollutants across scenarios and over the energy transition (See Figure 4, 5, and 6 in the main text). We believe this presents a clear way to compare the impact of different scenarios across different demographic groups. 2) We also capture air pollution changes across demographic groups by quantifying which scenario decreases the disparity the quickest (as seen in the above point). 3) We calculated the ratio between PM_{2.5} concentration reductions versus CO₂ reductions across race/ethnicity groups.

Another metric that would be quite valuable would be to report the ratio of PM to CO₂ according to different population categories over time so that we could observe which decarbonization policy leads to the greatest ton-for-ton improvement in health outcomes. Another metric would be to describe PM/MWh. The important thing that is missing overall is to describe the annual rate of change in reported measures.

To address the ton-for-ton improvement of health outcomes across different groups, we have created a ratio that compares reductions in population weighted PM_{2.5} concentrations to reductions in CO₂ (Table S-6). We see that scenario D (80% renewable energy by 2050) has the highest benefits across all race and ethnicity groups based on national reductions in CO₂. We have added discussion of these results into SI Section G.

[Lines 326-331 in main text] “We also investigate the reductions in population weighted PM_{2.5} concentration per billion metric tons of CO₂ reduced in 2050 across race and ethnicity groups (see SI Table S-6). We see that Scenario D has the highest ratio of reductions in PM_{2.5} concentration reductions per national reductions in CO₂ for all race and ethnicity groups. This means that we see the most benefit in health impacts (represented as reductions in PM_{2.5} concentrations) from reaching 80% renewable energy by 2050 compared to the reduction in CO₂ emissions.”

“Table S-6 shows the reductions in PM_{2.5} concentrations per billion metric tons of CO₂ reduced from 2020 to 2050. We see that Scenario D (80% renewable energy by 2050) has the highest ratio of reduction of PM_{2.5} to billion metric tons of CO₂ reduced. This means that we see the most benefit in health impacts (represented as reductions in PM_{2.5} concentrations) from reaching 80% renewable energy by 2050 per billion metric tons of CO₂ reduced.

Table S-6: Reductions in population weighted annual average PM_{2.5} concentration per billion metric tons of CO₂ reduced in 2050 (in µg-m⁻³ per billion metric tons).”

Race/Ethnicity Group	Scenario Emission Reduction Ratios (µg-m ⁻³ PM _{2.5} per billion metric tons CO ₂)							
	A	B	C	D	E	F	G	H
Black	0.526	0.327	0.347	0.568	0.523	0.527	0.525	0.526
Non-Latinx White	0.442	0.271	0.295	0.474	0.448	0.450	0.449	0.450
Asian	0.308	0.180	0.195	0.320	0.301	0.302	0.299	0.301
Latinx/Hispanic	0.269	0.144	0.169	0.291	0.279	0.280	0.276	0.278
Indigenous	0.290	0.154	0.183	0.311	0.303	0.305	0.304	0.305

Figure S-7 shows the ratio between PM_{2.5} emissions and generation (in kg/MWh). We see identical trends to Figure 2 in the main text, which shows national PM_{2.5} emissions for each scenario. Scenario E improves the best while Scenario B (US NDC carbon cap) does the second worst by 2050.

“Figure S-7: Ratio between PM_{2.5} emissions and generation (in kg/MWh). This shows the national emissions ratio of PM_{2.5} emissions per MWh generation. We see this is a similar trend across scenarios as to the national PM_{2.5} emissions in Figure 2.”

The authors do not have a table summarizing and describing the scenarios until it appears in the methods section. However, it is impossible to describe present the results without labeling the scenarios. The labels are not sufficient, and burden is placed on the reader to keep things straight based on the summary in the narrative. Hence, elements or all of the scenario table should be brought up in the manuscript to appear within the results section. This breaks “the rules” for manuscript organization but rules are meant to be broken when it serves purpose. The editor will need to weigh in on this.

We moved Table 1 to the beginning of the results to make the presentation of our results clearer. We also have written a more detailed text description of the scenarios in the results section to introduce the scenarios. Thank you for this insight.

Not until the very end of the paper is the reader reminded that costs of emissions mitigation may fall unevenly across the population. Indeed, as a share of income, the disadvantaged populations may bear a disproportionate cost which could have adverse health outcomes of comparable magnitude as examined here. I do not want to undermine the analysis of the benefits of air quality improvements, but this is more than a minor caveat and I believe it should be mentioned in the introduction.

This is a great point. We have added some discussion into the introduction to address this:

[Lines 41-43] “Even if income groups do have equal air quality concentrations, there still lies a disproportionate burden on lower income communities because they may not have the same access to health care as wealthier communities (Doty et al., 2020).”

14-17: The reader might infer that black or poor communities have higher pollutant concentrations during the transition than they do in the baseline.

We have updated the text to specify that it is in comparison to other groups. We have also updated our paper with population weighted analyses of racial groups (See Figure 4)

[Lines 17-22] “Black or poor communities are exposed to the highest concentrations of co-pollutants compared to other race and ethnicity groups and census tracts with lower poverty rates during the energy transition, until a national mandate requires >80% deployment of renewable or low-carbon technologies. However, they do see the greatest absolute decreases in air pollution exposure from the baseline in 2020.”

33: “aur”

We have corrected this.

57: “yet...” I am not sure the element of surprise that is implied by this word choice. It would be sufficient to directly state “and...”

Thank you, we have updated the wording here.

[Lines 62-64] “At a national level, air pollution is responsible for 100,000 to 200,000 excess deaths every year in the US and severe health effects, like lung heart, and brain diseases (Kampa & Castanas, 2008; Shi et al., 2021; Thakrar et al., 2020), and these effects are often greatest felt in minority communities (Ard, 2015; Jorgenson et al., 2020; Stuart et al., 2012; Tessum et al., 2021).”

85: Another reduced complexity paper that should be cited at the outset is a report that evaluates economy wide impacts and looks at the distribution of air quality benefits across demographic groups in a very similar way. See <https://www.rff.org/publications/reports/the-distribution-of-air-quality-healthbenefits-from-meeting-us-2030-climate-goals/>

Thank you for bringing this report to our attention. We have added this into our discussion of similar analyses:

[Lines] “Burtraw et al. (2022) use a capacity expansion model tied to a reduced complexity model to investigate the air pollution benefits of reaching climate goals by 2030 across counties and different demographic groups (Burtraw et al., 2022).”

93: “investigated researched”

This has been corrected.

208: “depositing” Air pollution is not deposited in communities, unless one is referring to acidification of the environment or other ecological outcomes. Air pollution concentrations

change and are transitory over geography. It would be better to write “...through the atmosphere affecting pollutant concentrations and causing health impacts.”

We have updated the paper to reflect this. For example, we have added:

[Lines 253-254] “Once emitted from the power plant, air pollution travels through the atmosphere affecting pollutant concentrations and causing health impacts (Heo et al., 2017).”

271-278: There may be deeper concerns than what is listed in this paragraph. Cumulative exposures combined with other community-level stressors point to different concentration-response relationships. See Spiller et al. Mortality Risk from PM_{2.5}: A Comparison of Modeling Approaches to Identify Disparities Across Ethnic Groups in Policy Outcomes.

Thank you for raising this point. We have added more discussion about the concerns around air pollution emissions. We have expanded our discussion of the disproportionate impacts on vulnerable communities, and included Spiller et al.

[Lines 259-360] “We also note that beyond this, cumulative exposure combined with community-level stressors will impact the health disparities among groups (Spiller et al., 2021).”

281: The legend should indicate that this is change in average PM concentration associated with changes in power generation.

Thank you for the indication of a clarity issue. We have replaced Figure 6 with population weighted average annual concentrations over 2020 – 2050 (Figure S-11 below). We note in the figure caption that changes in PM_{2.5} concentrations are associated with changes in power generation. We have added more mention of this connection in the main text.

“Figure S-11: Population weighted average annual PM_{2.5} concentration for all scenarios across income groups 2020 – 2050. The changes across PM_{2.5} concentration are associated with changes in the source of power generation.”

310: Black communities see the highest emissions. They also see the greatest reductions.

This is a very important point to address. We've added discussion about how the starting point of air pollution concentrations in different communities affects the benefits they receive. The starting points of air pollution stems from historic concentrations and inequities. This paper focuses on distributional justice, while this is a topic of recognitional justice. We find that the rate of change across race and ethnicity groups is very similar (seen in SI Figure S-17 below), so the starting point of concentrations is key to the future distribution of air pollution and who is burdened with more air pollution.

[Lines 307-311] “Based on these results, Black communities are at risk for higher PM_{2.5} concentrations and its associated health impacts in energy transitions but also see the largest absolute reductions in air pollution exposure (see SI Figure S-16). The percent change of PM_{2.5} concentration across race and ethnicity changes at the same rate within each scenario (see SI Figure S-17), indicating that the starting point of air pollution impacts future exposure throughout the energy transition.”

[Lines 415-420] “We also see that trends for who is exposed to higher concentrations of pollutants follow historical trends, so the distribution of air pollution over the energy transition using a least cost paradigm is dependent on the historical and current concentrations across demographic groups. Future work could investigate how historic inequities impact future air pollution distribution. This paper focuses on distributional justice, while this is a topic of recognitional justice.”

“Figure S-17: Percent change of population weighted PM_{2.5} concentration across race and ethnicity groups for each scenario over the energy transition 2020 – 2050.”

316: "...historical trends will exacerbate health impacts..." This may be true (see Spiller reference above) but cannot be asserted without a reference or supporting evidence.

We have added Spiller et al. (2021) to our references when discussing how historical trends in air quality could cause disparities across demographic groups.

We have removed this and discussed that future work can investigate how historical trends impact future air pollution disparities.

[Lines 418-420] "Future work could investigate how historic inequities impact future air pollution distribution. This paper focuses on distributional justice, while this is a topic of recognitional justice."

374: Does ReEDS achieve dynamic consistency? Is there a chance that an investment is made in, say, 2025 that is regrettable come 2030? This should be checked.

ReEDS uses a sequential solve method, which solves each year individually before moving to the next model year. Therefore, ReEDS has limited foresight into model input changes over time, like changes to the market or policies (Cohen et al., 2019). We have noted this in the text.

[Lines 480-482] "Because ReEDS solves sequentially, the model has limited foresight into model input changes over time, like changes to policies or the market (Cohen et al., 2019)."

401: What is the relevance of CO_{2e} as a metric given that only combustion related CO₂ is used to evaluate the climate policy?

Thank you for bringing this to our attention. We have clarified that the operating CO₂ emissions are CO₂ emissions, not CO_{2e}, in the paper. All instances of CO_{2eq} have been changed to CO₂.

407: I believe that Table 2 should be reported in pound/MWh. This will have much greater relevance for a broad community and make the paper more widely cited.

We have updated Table 2 to be in metric units of grams/kWh.

[Lines 503-507] "Table 2: Operating emission rates used in environmental sustainability analysis [in g/kWh]. See Table S-2 in SI for sources.*"

	CO ₂ [g/kWh]	NO _x [g/kWh]	SO ₂ [g/kWh]	PM _{2.5} ** [g/kWh]
Biopower	0	0	0.490	0.620
Solar photovoltaic (PV)	0	0	0	-
Concentrated solar power (CSP)	0	0	0	-
Onshore wind	0	0	0	-
Offshore wind	0	0	0	-
Nuclear	0	0	0	-

Natural gas combustion turbine (CT)	496	0.637	0.064	0.028
Natural gas combined cycle (CC)	337	0.058	0.015	0.019
Natural gas CCS	39.8	0.069	0.017	0.022
Hydropower	0	0	0	0
Geothermal	0	0	0	0
Oil-Gas-Steam	662	0.832	1.44	0.081
Coal	923	0.672	2.06	0.069 ¹
Coal IGCC	756	0.305	0.199	0.056 ¹
Coal CCS	97.0	0.392	0.256	0.072 ¹
Cofire	821	0.598	1.89	0.131
Battery storage	0	0	0	0
Pumped hydropower	0	0	0	0

References

- Ard, K. (2015). Trends in exposure to industrial air toxins for different racial and socioeconomic groups: A spatial and temporal examination of environmental inequality in the U.S. from 1995 to 2004. *Social Science Research, 53*, 375–390.
<https://doi.org/10.1016/J.SSRESEARCH.2015.06.019>
- Burtraw, D., Domeshek, M., Shih, J.-S., Villanueva, S., & Lambert, K. F. (2022). *The Distribution of Air Quality Health Benefits from Meeting US 2030 Climate Goals*.
- Cohen, S., Becker, J., Bielen, D., Brown, M., Cole, W., Eurek, K., Frazier, W., Frew, B., Gagnon, P., Ho, J., Jadun, P., Mai, T., Mowers, M., Murphy, C., Reimers, A., Richards, J., Ryan, N., Spyrou, E., Steinberg, D., ... Zwerling, M. (2019). *Regional Energy Deployment System (ReEDS) Model Documentation: Version 2018*.
<https://www.nrel.gov/docs/fy19osti/72023.pdf>.
- Dimanchev, E. G., Paltsev, S., Yuan, M., Rothenberg, D., Tessum, C. W., Marshall, J. D., & Selin, N. E. (2019). Health co-benefits of sub-national renewable energy policy in the US. *Environmental Research Letters, 14*(8), 085012. <https://doi.org/10.1088/1748-9326/AB31D9>
- Doty, M. M., Tikkanen, R. S., Fitzgerald, M., Fields, K., & Williams, R. D. (2020). Income-Related Inequality In Affordability And Access To Primary Care In Eleven High-Income Countries. *Https://Doi.Org/10.1377/Hlthaff.2020.01566, 40*(1), 113–120.
<https://doi.org/10.1377/HLTHAFF.2020.01566>
- Energy Innovation Policy & Technology LLC. (n.d.). *United States | Energy Policy Solutions*. Retrieved November 16, 2020, from <https://us.energypolicy.solutions/scenarios/home>
- Heo, J., Adams, P. J., & Gao, H. O. (2017). Public health costs accounting of inorganic PM2.5 pollution in metropolitan areas of the United States using a risk-based source-receptor model. *Environment International, 106*, 119–126.
<https://doi.org/10.1016/J.ENVINT.2017.06.006>
- Jbaily, A., Zhou, X., Liu, J., Lee, T. H., Kamareddine, L., Verguet, S., & Dominici, F. (2022). Air pollution exposure disparities across US population and income groups. *Nature 2022 601:7892, 601*(7892), 228–233. <https://doi.org/10.1038/s41586-021-04190-y>
- Jorgenson, A. K., Hill, T. D., Clark, B., Thombs, R. P., Ore, P., Balistreri, K. S., & Givens, J. E. (2020). Power, proximity, and physiology: does income inequality and racial composition amplify the impacts of air pollution on life expectancy in the United States? *Environmental Research Letters, 15*(2), 024013. <https://doi.org/10.1088/1748-9326/AB6789>
- Kampa, M., & Castanas, E. (2008). Human health effects of air pollution. In *Environmental Pollution* (Vol. 151, Issue 2, pp. 362–367). Elsevier.
<https://doi.org/10.1016/j.envpol.2007.06.012>
- Lane, H. M., Morello-Frosch, R., Marshall, J. D., & Apte, J. S. (2022). Historical Redlining Is Associated with Present-Day Air Pollution Disparities in U.S. Cities. *Environmental Science and Technology Letters, 9*(4), 345–350.
https://doi.org/10.1021/ACS.ESTLETT.1C01012/ASSET/IMAGES/LARGE/EZ1C01012_0002.JPEG
- Li, M., Hilpert, M., Goldsmith, J., Brooks, J. L., Shearston, J. A., Chillrud, S. N., Ali, T., Umans, J. G., Best, L. G., Yracheta, J., van Donkelaar, A., Martin, R. v., Navas-Acien, A., &

- Kioumourtzoglou, M.-A. (2022). Air Pollution in American Indian Versus Non–American Indian Communities, 2000–2018. *American Journal of Public Health, 112*(4), 615–623. <https://doi.org/10.2105/AJPH.2021.306650>
- Luo, Q., Johnson, J. X., & Garcia-Menendez, F. (2021). Reducing human health impacts from power sector emissions with redispatch and energy storage. *Environmental Research: Infrastructure and Sustainability, 1*(2), 025009. <https://doi.org/10.1088/2634-4505/AC20B3>
- Mayfield, E. N. (2022). Phasing out coal power plants based on cumulative air pollution impact and equity objectives in net zero energy system transitions. *Environmental Research: Infrastructure and Sustainability, 2*(2), 021004. <https://doi.org/10.1088/2634-4505/AC70F6>
- Nock, D., Levin, T., & Baker, E. (2020). Changing the policy paradigm: A benefit maximization approach to electricity planning in developing countries. *Applied Energy, 264*, 114583. <https://doi.org/10.1016/j.apenergy.2020.114583>
- Prener, C. (n.d.). *Areal Weighted Interpolation*. Retrieved December 17, 2021, from <https://cran.r-project.org/web/packages/areal/vignettes/areal-weighted-interpolation.html>
- S Thind, M. P., Tessum, C. W., Azevedo, L., & Marshall, J. D. (2019). *Fine Particulate Air Pollution from Electricity Generation in the US: Health Impacts by Race, Income, and Geography*. <https://doi.org/10.1021/acs.est.9b02527>
- Saari, R. K., Selin, N. E., Rausch, S., & Thompson, T. M. (2015). A self-consistent method to assess air quality co-benefits from U.S. climate policies. *Journal of the Air and Waste Management Association, 65*(1), 74–89. <https://doi.org/10.1080/10962247.2014.959139>
- Sasse, J. P., & Trutnevyte, E. (2020). Regional impacts of electricity system transition in Central Europe until 2035. *Nature Communications, 11*(1), 1–14. <https://doi.org/10.1038/s41467-020-18812-y>
- Sergi, B. J., Adams, P. J., Muller, N. Z., Robinson, A. L., Davis, S. J., Marshall, J. D., & Azevedo, I. L. (2020). Optimizing Emissions Reductions from the U.S. Power Sector for Climate and Health Benefits. *Environmental Science & Technology, 54*(12), 7513–7523. <https://doi.org/10.1021/ACS.EST.9B06936>
- Shi, L., Steenland, K., Li, H., Liu, P., Zhang, Y., Lyles, R. H., Requia, W. J., Ilango, S. D., Chang, H. H., Wingo, T., Weber, R. J., & Schwartz, J. (2021). A national cohort study (2000–2018) of long-term air pollution exposure and incident dementia in older adults in the United States. *Nature Communications 2021 12:1, 12*(1), 1–9. <https://doi.org/10.1038/s41467-021-27049-2>
- Spiller, E., Proville, J., Roy, A., & Muller, N. Z. (2021). Mortality Risk from PM2.5: A Comparison of Modeling Approaches to Identify Disparities across Racial/Ethnic Groups in Policy Outcomes. *Environmental Health Perspectives, 129*(12). <https://doi.org/10.1289/EHP9001>
- Stuart, A. L., Mudhasakul, S., & Sriwatanapongse, W. (2012). The Social Distribution of Neighborhood-Scale Air Pollution and Monitoring Protection. [Http://Dx.Doi.Org/10.3155/1047-3289.59.5.591](http://Dx.Doi.Org/10.3155/1047-3289.59.5.591), 59(5), 591–602. <https://doi.org/10.3155/1047-3289.59.5.591>

- Tarekegne, B. W., Kazmierczuk, K., & O'neil, R. S. (2021). *Coal-dependent Communities in Transition: Identifying Best Practices to Ensure Equitable Outcomes*.
<https://www.ntis.gov/about>
- Tessum, C. W., Hill, J. D., & Marshall, J. D. (2017). InMAP: A model for air pollution interventions. *PLOS ONE*, *12*(4), e0176131.
<https://doi.org/10.1371/JOURNAL.PONE.0176131>
- Tessum, C. W., Paoella, D. A., Chambliss, S. E., Apte, J. S., Hill, J. D., & Marshall, J. D. (2021). PM2.5 pollutants disproportionately and systemically affect people of color in the United States. *Science Advances*, *7*(18), 4491–4519.
https://doi.org/10.1126/SCIADV.ABF4491/SUPPL_FILE/ABF4491_SM.PDF
- Thakrar, S. K., Balasubramanian, S., Adams, P. J., Azevedo, I. M. L., Muller, N. Z., Pandis, S. N., Polasky, S., Pope, C. A., Robinson, A. L., Apte, J. S., Tessum, C. W., Marshall, J. D., & Hill, J. D. (2020). Reducing Mortality from Air Pollution in the United States by Targeting Specific Emission Sources. *Environmental Science and Technology Letters*, *7*(9), 639–645.
https://doi.org/10.1021/ACS.ESTLETT.0C00424/SUPPL_FILE/EZ0C00424_SI_002.ZIP
- Thompson, T. M., Rausch, S., Saari, R. K., & Selin, N. E. (2014). A systems approach to evaluating the air quality co-benefits of US carbon policies. *Nature Climate Change* *2014 4:10*, *4*(10), 917–923. <https://doi.org/10.1038/nclimate2342>
- Zhang, Y., Smith, S. J., Bowden, J. H., Adelman, Z., & Jason West, J. (2017). Co-benefits of global, domestic, and sectoral greenhouse gas mitigation for US air quality and human health in 2050. *Environmental Research Letters*, *12*(11), 114033. <https://doi.org/10.1088/1748-9326/AA8F76>

REVIEWER COMMENTS

Reviewer 1 has not left additional notes to the Author, but recommended the paper for publication to the Editors.

Reviewer #2 (Remarks to the Author):

The authors have thoroughly responded to all of my concerns. I believe the edited version of the manuscript is greatly improved and would make an excellent contribution to Nature Communications.

Reviewer #3 (Remarks to the Author):

The authors have executed changes in response to comments from the three reviewers that strengthen the paper and especially that help communicate the main findings. I have the following additional observations and comments.

1. I appreciate the newly displayed results in S6 and S7. I think the lessons from this material deserve some attention in the main text, perhaps with a statement that refers the reader to supplemental material. I feel this is important because this information offers some of the most tangible policy advice in the paper.

Moreover, I would like to see the information from table S6 displayed in a line graph over time. The fundamental question for a policy maker is how to make the most progress the most quickly in arresting disparities while also achieving related climate goals. The paper points to final outcomes (targets) but not to policy pathways, which might imply a different recommendation.

My concern about the information in the paper as currently presented is that the takeaway is that disparities persist until at least 80% renewables are in the mix. It makes sense that the ratio of air benefits to CO₂ reductions is greatest for a policy that achieves just 80%. However, really what the policy maker wants to know (or may want to know hopefully) is which policy pathway aiming at net zero achieves the greatest emissions reductions per ton CO₂ removed along that pathway. I think that may provide a different answer than what is reflected in table S6.

2. The paper models the power sector but major changes from electrification are expected outside the power sector. In making this point incidentally, the authors might offer an observation of the demographic characteristics of the populations affected by emissions from transportation (proximity to roadways) and buildings (quality of the housing stock). At line 337 the relevance of the transportation sector is mentioned. More could be said. I invite the authors to editorialize briefly about the implications of electrification. This begs the question of whether additional investments to achieve net zero in the electricity sector might raise electricity prices or divert public support from efforts to accelerate electrification, which brings a different portfolio of benefits. The authors should comment on this, or at least queue this up as a central question for policy makers.

3. The authors report that the magnitude of change across race and ethnicity varies, the percent change of emissions across race and ethnicity groups are almost identical within a scenario, which indicates that the starting point of air pollution concentrations is a significant indicator of future inequalities. That is helpful and clear. It might be helpful to reflect on differences across regions in the same context. Does the population distribution by race by region explain greater exposure for the black population (and non-Latinx whites)? The authors indicate already this could be true, due for example to the distribution of the population in rural areas. Does this regional distributional difference fit the same framework as the strength of the paper which looks at effects within a region?

4. For Table 2, I believe the MWh denominator is net contribution to the grid after accounting for parasitic loss of post combustion controls. This should be stated with the table for absolute clarity.

Air pollution disparities and equality assessments of US national decarbonization strategies

Response to reviewer comments*

***Reviewer comments are indicated in black, our responses in blue, and our text additions in red. References are at the end of the document.**

Reviewer 3

1. I appreciate the newly displayed results in S6 and S7. I think the lessons from this material deserve some attention in the main text, perhaps with a statement that refers the reader to supplemental material. I feel this is important because this information offers some of the most tangible policy advice in the paper.

Thank you for the positive comment. To highlight the value of figure S6 and S7 we have added some discussion on Figure S-7 and Table S-6 in the main text.

[Lines 228-235] “We also investigate the national ratio of PM_{2.5} emissions per megawatt-hour of annual national generation to obtain national emissions rates across the decarbonization scenarios (SI Figure S-7). We find that the 100% renewable energy scenarios have the lowest PM_{2.5} emissions per national annual megawatt-hour generation by their mandate year. Interestingly we see that in the absence of a strict renewable energy mandate, the 1.5°C decarbonization pathway (Scenario C) often has the lowest or second lowest ratio over the entire modeling horizon from applying an aggressive carbon cap. This most likely stems from Scenario C (1.5°C decarbonization pathway) retiring the entire coal fleet in the same time period as Scenario E (100% RE by 2035).”

[Lines 325-333] “We also investigate the change in PM_{2.5} with respect to the change in CO₂ in 2050 across race and ethnicity groups (see SI Table S-6). If the ratio is less than one, it indicates that CO₂ is reduced faster than PM_{2.5}. We can use this ratio to understand the rate of PM_{2.5} reductions across different decarbonization pathways and racial groups. However, the ratio does not capture the magnitude of reductions, which could impact decision-making. We see that Scenario D (80% RE by 2050) has the highest ratio of reductions in PM_{2.5} concentration reductions per national reductions in CO₂ for all race and ethnicity groups from 2020 to 2050. This indicates that while Scenario D does not reach zero CO₂ or PM_{2.5} emissions, it reduces local pollutants at a faster rate per CO₂ reduction than other scenarios.”

Moreover, I would like to see the information from table S6 displayed in a line graph over time. The fundamental question for a policy maker is how to make the most progress the most quickly in arresting disparities while also achieving related climate goals. The paper points to final outcomes (targets) but not to policy pathways, which might imply a different recommendation.

We have displayed the information from Table S-6 in two line graphs over time (as seen below): split out by scenario and by race or ethnicity group. The graphs show the ratio between PM_{2.5}

concentration reductions and CO₂ reductions over about 5-year time spans. We equate a higher ratio to more benefits because a higher ratio indicates that PM_{2.5} is being reduced at greater levels than CO₂. This places an emphasis on reductions in co-pollutants in the pathway to decarbonizing, which is important to reducing health impacts on communities over the energy transition.

While we find the ratio over time interesting, we note that there are counterintuitive interpretations of this graph. For example, in 2040, and in some cases in 2026, we see the ratio dip below zero. This indicates that one of the emissions is increasing while the other is decreasing. Another thing we have found is that in between 2020 and 2050, sometimes both emissions increase, which still produces a positive ratio. Thus, having a graph titled ratio of emissions reductions would be misleading. While we agree that ratios can highlight policy pathways over time, we have elected to retain the table as the main discussion of ratios because our paper is focused on evaluating policy targets. We updated SI Table S-6 to make it easier to identify the most impacted community groups.

Reviewer Response Figure: A ratio of reductions in PM_{2.5} per reductions in CO₂ across scenarios, separated by race or ethnicity. Note that the graph does not include scenario E (100% RE 2035) due to CO₂ emissions reaching essentially 0 and thus the ratio communicated infinite benefits in 2040 and 2050. Thus, while scenario E has the highest ratio in 2040 and 2050, we highlight that if E is not chosen as a policy pathway, then Scenario B (US NDC carbon cap) in 2050 and Scenario D (80% RE 2050) consistently present the greatest reduction in PM_{2.5} per CO₂ reduced across the 30-year time period. It is important to note that while these scenarios show

the greatest benefits with respect to this ratio, they are two scenarios that continue fossil fuel generation without CCS through 2050.

Reviewer Response Figure: A ratio of reductions in PM_{2.5} per reductions in CO₂ across scenarios, separated by race or ethnicity. We include Scenario E in this figure, and it shows ‘infinite benefits’ in the years 2040 and 2050 due to almost zero CO₂ emissions and reductions.

My concern about the information in the paper as currently presented is that the takeaway is that disparities persist until at least 80% renewables are in the mix. It makes sense that the ratio of air benefits to CO₂ reductions is greatest for a policy that achieves just 80%. However, really what the policy maker wants to know (or may want to know hopefully) is which policy pathway aiming at net zero achieves the greatest emissions reductions per ton CO₂ removed along that pathway. I think that may provide a different answer than what is reflected in table S6.

The 80% renewable by 2050 scenario (Scenario D) has the highest reductions in air pollution with respect to CO₂ reductions when looking over the entire timescale of 2020 to 2050, this can be seen in Table S-6. However, with the new plots, we see that over Scenario B (US NDC carbon cap) performs the best in 2050, Scenario E provides ‘infinite benefits’ in 2040 and 2050 due to its negligible emission, and Scenario D provides the most consistent benefits to race and ethnicity groups over the energy transition since its ratio over the time period is the most horizontally linear. In the paper, we have highlighted the total emission changes over the pathway in Figure 2. We believe this addresses the goal of how benefits are achieved over decarbonization pathways. We also have found that highlighting total emissions over the pathway is more intuitive than using ratios.

2. The paper models the power sector but major changes from electrification are expected outside the power sector. In making this point incidentally, the authors might offer an observation of the demographic characteristics of the populations affected by emissions from transportation (proximity to roadways) and buildings (quality of the housing stock). At line 337 the relevance of the transportation sector is mentioned. More could be said. I invite the authors to editorialize briefly about the implications of electrification. This begs the question of whether additional investments to achieve net zero in the electricity sector might raise electricity prices or divert public support from efforts to accelerate electrification, which brings a different portfolio of benefits. The authors should comment on this, or at least queue this up as a central question for policy makers.

We have added discussion about other sectors that are not captured in our analysis in the Limitations section. While the points mentioned here are valid, investigating the demographic characteristics of people impacted by the transportation and residential sectors is out of the scope of our paper. However, we have noted that this is an important future step of this analysis.

[Lines 602-608] “While there are many possible impacts from decarbonization efforts in countries, we note that our analysis only investigates the air pollution impacts within the electricity sector in a developed country. Investigating decarbonization impacts from other

sectors, like transportation, residential housing, and industry is an important next step in the understanding the full extent of inequities in the energy system and how they will be impacted by the energy transition. Deployment of low carbon technology will also have different implications in emerging countries who wish to expand their electricity systems.”

3. The authors report that the magnitude of change across race and ethnicity varies, the percent change of emissions across race and ethnicity groups are almost identical within a scenario, which indicates that the starting point of air pollution concentrations is a significant indicator of future inequalities. That is helpful and clear. It might be helpful to reflect on differences across regions in the same context. Does the population distribution by race by region explain greater exposure for the black population (and non-Latinx whites)? The authors indicate already this could be true, due for example to the distribution of the population in rural areas. Does this regional distributional difference fit the same framework as the strength of the paper which looks at effects within a region?

To investigate the impact of regional distribution of different race groups, we map the percent of Black and non-Latinx white population within a census tract and show the distribution of air pollution in the Scenario A (the base case) in 2020 and 2050 in Figure S-20 below. We find that regions with census tracts of over 95% white have some of the highest concentrations of PM_{2.5} in 2020. By 2050 in the Base Case, there are still high concentrations of PM_{2.5} in Ohio. We see census tracts that are over 20% Black are mainly in along the Gulf of Mexico, through Florida, and up the East Coast. The regional distribution of air pollution from the electricity sector has higher concentrations in these regions. Therefore, the population distribution of race and ethnicity groups impacts their exposure to co-pollutants from power plants. We have added discussion of this in the main text.

[Lines 310-313] “When investigating the regional distribution of race and ethnicity groups across the US (SI Figure S-20), we find that regions with census tracts over 95% non-Latinx white and census tracts that are >10% Black have some of the highest PM_{2.5} concentrations.”

“Figure S-20: Regional distribution of air pollution and racial/ethnic groups. The distribution of racial groups includes the (a) Black and (c) non-Latinx white communities by census tract. The distribution of air pollution in the Base Case (Scenario A) are shown for the years (b) 2020 and (d) 2050. We show the Black and white racial groups because these communities are exposed to the highest concentrations of PM_{2.5} from the electricity sector across the entire modeling horizon in our analysis.”

4. For Table 2, I believe the MWh denominator is net contribution to the grid after accounting for parasitic loss of post combustion controls. This should be stated with the table for absolute clarity.

Thank you for pointing this out. This distinction will depend on the calculation method used in the emissions assessments. We have added a statement that highlights the sources for our emissions rates.

REVIEWERS' COMMENTS

Reviewer #3 (Remarks to the Author):

The authors have responded thoughtfully to my most recent comments and questions. The paper is in great shape.